# Towards Harmless Rawlsian Fairness Regardless of Demographic Prior

**Xuanqian Wang**[1*]    **Jing Li**[2,3†]    **Ivor W. Tsang**[2,3,4]    **Yew-Soon Ong**[2,3,4]

[1]School of Computer Science and Engineering, Beihang University, China
[2]Institute of High-Performance Computing, Agency for Science, Technology and Research, Singapore
[3]Centre for Frontier AI Research, Agency for Science, Technology and Research, Singapore
[4] College of Computing and Data Science, Nanyang Technological University, Singapore
wwxxqq@buaa.edu.cn  {kyle.jingli, ivor.tsang}@gmail.com  asysong@ntu.edu.sg

## Abstract

Due to privacy and security concerns, recent advancements in group fairness advocate for model training regardless of demographic information. However, most methods still require prior knowledge of demographics. In this study, we explore the potential for achieving fairness without compromising its utility when no prior demographics are provided to the training set, namely *harmless Rawlsian fairness*. We ascertain that such a fairness requirement with no prior demographic information essential promotes training losses to exhibit a Dirac delta distribution. To this end, we propose a simple but effective method named VFair to minimize the variance of training losses inside the optimal set of empirical losses. This problem is then optimized by a tailored dynamic update approach that operates in both loss and gradient dimensions, directing the model towards relatively fairer solutions while preserving its intact utility. Our experimental findings indicate that regression tasks, which are relatively unexplored from literature, can achieve significant fairness improvement through VFair regardless of any prior, whereas classification tasks usually do not because of their quantized utility measurements. The implementation of our method is publicly available at `https://github.com/wxqpxw/VFair`.

## 1 Introduction

Fairness in machine learning has gained significant attention owing to its multifaceted ethical implications and its far-reaching potential to shape and influence various aspects of society [1, 2, 3]. In high-stakes decision-making domains, algorithms that merely prioritize model utility may yield biased models, resulting in unintentional discriminatory outcomes concerning factors such as gender and race. Group fairness, a.k.a. statistical fairness [4], addresses this issue by explicitly encouraging the model behavior to be independent of group indicators, such as disparate impact [5], or equalized odds [6]. However, with increasing privacy concerns applied in practical situations, sensitive attributes are not accessible which raises a new challenge for fairness learning.

According to literature, numerous efforts have been directed towards achieving fairness regardless of demographic information, which can be mainly categorized into two branches. One branch is to employ proxy-sensitive attributes [7, 8, 9, 10]. These works assume that estimated or selected attributes are correlated with the actual sensitive attributes and thus can serve as a proxy of potential biases. The other branch follows Rawlsian fairness [11], which focuses on reducing the disparity in group utility. Unlike the former branch, the group utility here is predefined and centered, and thus it is

---

[*]Work done during an internship at A*STAR.
[†]Corresponding author.

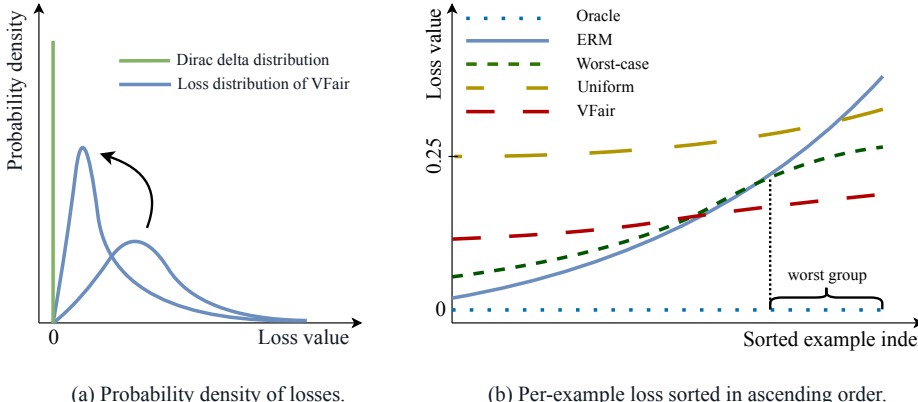

(a) Probability density of losses.    (b) Per-example loss sorted in ascending order.

Figure 1: Illustration of our idea through different forms of loss curves.

typically not meaningful to test learned models on other group fairness metrics. Worst-case fairness methods [12, 13], belonging to Rawlsian fairness, commonly leverage a prior about the worst-off group's size to identify under-represented members and prioritize the utility of the approximate worst-off group. Such a taxonomy overlooks the differences in tasks, as the majority of aforementioned fairness approaches are designed for classification tasks. In applications where discrete outcomes (e.g. binary decisions) provide insufficient information, there is a crucial need for fair regressors. As relatively few discussions [14, 15] exist for fair regression tasks, our work bridges the gap by incorporating regression tasks into a general predictive loss under Rawlsian fairness.

The trade-off nature between model utility and fairness has emerged as a subject of dynamic discourse [16, 17, 18]. Worst-case fairness methods which inherently prioritize the worst-off group's utility often come at the expense of the overall utility [13, 19]. In this work, we focus on scenarios where no prior demographic information is provided, aligning the ingredients with the standard training setup, and then advocate for a primary problem (harmless Rawlsian fairness):

*Regardless of demographic prior, to what extent can we improve Rawlsian fairness without hurting the model's overall utility?*

This problem is particularly important in utility-intensive scenarios [20], and we investigate it in both classification and regression tasks to answer the question.

**Our idea.** We approach the problem from a novel perspective. The crux of the desired fairness lies in pursuing minimum group utility disparity across all groups. Since during the training phase, we are not aware of what the actual sensitive attributes are used for test data, the safest way is to ensure every possible disjoint group of training data has the same utility. To this end, we expect the training loss for each individual example to be very close, meaning that the loss variable approaches a Dirac delta distribution. As shown in Fig. 1 (a). The Dirac delta distribution essentially represents an Oracle model, where all the loss values are concentrated at zero, resulting in both a mean and variance of 0. In the distribution view, the motivation of our method which is dubbed as VFair, is to approximate this ideal by minimizing both the mean and variance of the training losses. Fig. 1 (b) also shows the comparison between VFair and other methods, using a regression task as an example. Oracle denotes models with unlimited capability that make predictions with zero error. Empirical Risk Minimization (ERM) refers to models without any fairness design. Worst-case represents fairness methods that require the prior of the worst-off group (e.g., lower bound of partition ratios). Uniform model, initially introduced by [12], represents a model that performs equally poorly across all groups on classification tasks. Here, we extend it to regression. In a simplified logistic regression task that applies Mean Squared Error (MSE) loss with targets of 0 or 1, a "uniform regressor" predicts values close to 0.5 for each example, resulting in losses close to 0.25, as indicated by the yellow dashed line. As depicted by Fig. 1 (b), we expect VFair to exhibit the following two properties.

(1) VFair achieves a more flattened loss curve compared to ERM and Worst-case. A flattened curve indicates similar losses for each example, indicating a fairer solution for unknown group partitions. (2) VFair maintains an area under the curve comparable to that of ERM, reflecting the overall model utility. Since a flattened curve may deteriorate into a uniform model that significantly sacrifices overall utility, VFair prioritizes keeping the overall average loss at a low value.

Statistically, our main idea can be understood as minimizing the loss distribution's second moment (e.g., loss variance) while not increasing its first moment. By developing a dynamic approach operated at both the loss and gradient levels, our idea is proven feasible and effective.

**Contributions.** The key contributions of this research can be outlined as follows.

• We introduce the setting of harmless Rawlsian fairness regardless of demographic prior in both classification and regression tasks. To well position this setting, we also discuss its connections with Worst-case fairness and harmless fairness from the view of variance reduction and re-weighting.

• We advocate that minimizing the variance of prediction losses is a straightforward yet effective fairness proxy. By incorporating it as a secondary objective, the overall model performance can remain uncompromised.

• We develop a dynamic approach for conducting harmless updates, which is operated at both the loss and gradient levels, guiding the model towards a fair solution without compromising utility.

• We analyze the difference between fair regression and classification tasks, and experimentally demonstrate that, regardless of any prior, harmless Rawlsian fairness is achievable in regression tasks but unfortunately not in classification tasks.

## 2   Related work

**Worst-case fairness without demographics.** In alignment with the Rawlsian fairness principle, a sequence of studies has followed the Worst-case scheme, which focuses on improving the performance of the worst-off group without full demographics. DRO [13] identified the worst-off group members by a lower bound for the minimal group ratio. The behind insight is that examples yielding larger losses are more likely sampled from an underprivileged group and thus should be up-weighted, which inherits the fairness strategy for handling group imbalance [21, 22]. Similarly, [12] also considered training a fair model with a given ratio of the protected group and connected such a fairness learning setting with the subgroup robustness problem [23]. In contrast to these studies, ARL [24] introduced the concept of Computational-Identifiability to enhance the Worst-case scheme. ARL presented an adversarial re-weighting method to identify the worst-off group in the computational-identifiable region without relying on any demographic prior. This embodies the genuine essence of achieving fairness without demographics and is closest to our setting.

**Harmless fairness.** In utility-intensive scenarios, a fair model is meaningful only when it preserves good utility. Basically, these works engaged in discussing the extent to which fairness can be achieved without compromising model utility. Some [25, 26] searched for the so-called minimax Pareto fair optimality for off-the-shelf binary attributes and then upgraded their method to the multi-value attribute cases with only side information about group size [12]. A pre-processing method [20] accomplished cost-free fairness through re-weighting training examples based on both fairness-related measures and predictive utility on a validation set. Based on the concept of Rashomon Effect, [27] achieved fairness from good-utility models under selective labels through a constrained optimization perspective, needing a proper upper bound for the average loss. The same fairness notation also applies to regression task [15], where the prediction error of protected groups remains below some predefined threshold, and the fairness-accuracy frontier is experimentally achieved. Notably, these works more or less require direct or implicit demographic information and cannot adapt to our problem setting. A dynamic barrier gradient descent algorithm [28] was recently introduced which allows models to prioritize must-satisfactory constraints. Inspired by this, we conceptualize harmless fairness within a similar framework, enabling us to move beyond a utility-only solution and obtain a fairer model that can narrow the utility gaps among possible data partitions.

## 3   VFair methodology

### 3.1   Problem setup

Consider a supervised learning problem from input space $\mathcal{X}$ to a label space $\mathcal{Y}$, with training set $\{z_i\}_{i=1}^N$, where $z_i = (x_i, y_i) \in \mathcal{X} \times \mathcal{Y}$. For a model parameterized by $\theta \in \Theta$ and a training point $z^{\#}$,

---

[#]Throughout this paper, random variables are represented with lowercase letters unless otherwise specified.

let $\ell(z; \theta)$ be the associated loss. Suppose for each $z_i$, there exists a sensitive attribute $s_i \in \mathcal{S}$. Thus a $K$-value sensitive attribute $s$ will naturally partition data into $K$ disjoint groups. Such sensitive attributes are not observed during model training but can be accessible for fairness testing. Following the principle of Rawlsian fairness, the utility disparity over groups is used as a fairness evaluation metric. For example, in classification tasks, denoting $u_k$ as the classification accuracy of the $k$-th group, we can define the maximum utility disparity, i.e., $\mathrm{MUD} = \max_{i,j \in [K]} (u_i - u_j)$, as a proper fairness metric. More metrics will be introduced in Section 4.1, and the same applies to regression tasks. The fundamental objective of this work is to develop a model that minimizes group utility disparity [24, 12, 15] to the greatest extent possible while maintaining the overall predictive utility (compared to ERM) regardless of demographic prior.

### 3.2 Fairness via minimizing variance of losses

An ERM model may exhibit variable predictive utility across different groups. Conventionally, a fair counterpart is achievable by properly incorporating the objective of minimizing group utility disparity (e.g., MUD), which is however not applicable when demographics are not accessible at training stages. Intuitively, a predictive model that can be fair for any arbitrary partitions on the test set implies that the loss of each training example should be close to each other, exhibiting a Dirac delta distribution. A compelling piece of evidence is that an Oracle model, as depicted in Fig. 1 (b), ensures that each individual loss $\ell(z; \theta)$ is sufficiently small, resulting in no disparity, i.e., $\mathrm{MUD} = 0$. This case suggests that group fairness can be instance-wise characterized and hence bypasses the unobserved sensitive attributes. We present this insight by the following proposition.

**Proposition 1.** $u \perp s$ *holds for any $s$ that splits data into a number of groups, if and only if the loss $\ell$ is (approximately) independent of the training example $z$, i.e., $\ell \perp z$.*

The proof of Proposition 1 is left to Appendix A, and the approximation arises from the quantization of utility metrics, e.g., classification accuracy.

To achieve such a Dirac delta distribution, several fairness objectives can be adopted. We defer the discussion to Appendix B and conclude that applying the variance of losses as a fairness objective is simple yet efficient. Intuitively, the small variance does encourage the loss to be invariant of input. Suppose that we intend to achieve a small MUD through minimizing the maximum group utility disparity, denoted by $\ell_{\mathrm{MUD}}$. The following proposition shows that standard deviation of training losses essentially serves as a useful proxy.

**Theorem 1.** $\forall s \in \mathcal{S}, \forall \theta \in \Theta, \ell_{MUD} \leq C \sqrt{\mathbb{V}_z[\ell(z; \theta)]}$, *where $C$ is a constant.*

Appendix B.1 gives the proof of Theorem 1. Although $\ell_{\mathrm{MUD}}$ is upper-bounded in the form of standard deviation as stated in Theorem 1, we term it "variance" for convenience in statements where it does not introduce ambiguity. So far, we connect Rawlsian fairness with the variance of training losses, without using any prior of demographics.

### 3.3 Objective formulation

Notably, solely penalizing the variance of losses will not necessarily decrease the expectation of losses, leading to the emergence of a uniform model [12]. Thus, to improve fairness without compromising the overall model utility, our full objective is formulated as follows:

$$\min_{\theta \in \Theta} \sqrt{\mathbb{V}_z[\ell(z; \theta)]} \quad s.t. \ \mathbb{E}_z[\ell(z; \theta)] \leq \delta, \tag{1}$$

where $\delta$ controls how much we can tolerate the harm on the overall predictive utility, and the sense that $\delta = \inf_{\theta \in \Theta} \mathbb{E}_z[\ell(z; \theta)]$ suggests a zero-tolerance. In particular, we fold in any regularizers into $\ell(\cdot)$ to make our method easily adapt to specific scenarios. The empirical risk of Eq. 1 is written as

$$\min_{\theta \in \Theta} \underbrace{\sqrt{\frac{1}{N} \sum_{i=1}^{N} (\ell(z_i; \theta) - \hat{\mu}(\theta))^2}}_{\hat{\sigma}(\theta)} \quad s.t. \ \underbrace{\frac{1}{N} \sum_{i=1}^{N} \ell(z_i; \theta)}_{\hat{\mu}(\theta)} \leq \delta. \tag{2}$$

We use $\hat{\mu}(\theta)$ and $\hat{\sigma}(\theta)$ to denote the primary and secondary objectives, respectively. Since we minimize $\hat{\sigma}(\theta)$ inside the optimal set of minimizing $\hat{\mu}(\theta)$, the eventually learned model is viewed to be harmlessly fair with regard to the overall performance.

Note that the objective of Eq. 1 looks similar to variance-bias research [29, 30]. Following Bennett's inequality, the expected risk can be upper bounded by the empirical risk plus a variance-related term with a high probability:

$$\mathbb{E}_z[\ell(z;\theta)] \leq \frac{1}{N}\sum_{i=1}^{N}\ell(z;\theta) + C_1\sqrt{\frac{\mathbb{V}_z[\ell(z;\theta)]}{N}} + \frac{C_2}{N}, \tag{3}$$

where $C_1$ and $C_2$ are some constants which reflect the confidence guarantee. We emphasize Eq. 3 is apparently distinct from our fairness motivation. As derivated in [30], the right-hand side of Eq. 3 can be well approximately by a robust regularized risk, a.k.a. DRO's objective [31],

$$\min_{\theta\in\Theta}\inf_{\eta\in\mathbb{R}}\left\{F(\theta;\eta) := C\left(\mathbb{E}_z\left[[\ell(z;\theta) - \eta]_+^2\right]\right)^{\frac{1}{2}} + \eta\right\}, \tag{4}$$

where $C = \left(2(1/\alpha_{min} - 1)^2 + 1\right)^{1/2}$ and $\alpha_{min}$ is a bound of the worst-off group's ratio. Given an $\eta^t$ which is the optimal solution of the $t$-th inner optimization but also happens to be close to the mean loss, i.e., $\eta^t \approx \hat{\mu}(\theta^t)$, Eq. 4 can be viewed as penalizing the upper semi-variance of training loss. This observation connects Worst-case fairness with variance penalization from a new aspect. Although focusing on the variance of training losses in our method as well, we penalize it inside the optimal set of empirical losses. Our method is eventually capable of achieving harmless fairness.

### 3.4 Harmless fairness update

Directly calculating the optimal set of $\hat{\mu}(\theta) \leq \delta$ in Eq. 2 can be very expensive. A common approach is to consider an unconstrained form of Eq. 2, i.e., Lagrangian function, which however needs to not only specify a proper $\delta$ beforehand but also optimize the Lagrange multiplier to satisfy the constraint best. Besides, the constrained form of Eq. 2 makes our task different from traditional multi-objective optimization tasks. Recognizing that such re-balancing between two loss terms essentially operates on gradients, in a manner analogous to the approach outlined by [28], we consider the following gradient update scheme,

$$\theta^{t+1} \leftarrow \theta^t - \gamma^t\left(\lambda^t\nabla\hat{\mu}(\theta^t) + \nabla\hat{\sigma}(\theta^t)\right), \tag{5}$$

where $\gamma^t$ is a step size and $\lambda^t(\lambda^t \geq 0)$ is the dynamic coefficient we aim to get. For simplicity, we omit the time stamp $t$ and model variable $\theta$ when it is not necessary to emphasize them. Now we provide how do we dynamically adjust $\lambda$.

**Gradient view.** The idea of designing $\lambda$ is to keep decreasing $\hat{\mu}$ when the constraint is not met, meaning that the combined gradient should never hurt the descent of $\hat{\mu}$. As depicted in Fig. 2 (a), if the gradients $\nabla\hat{\sigma}$ and $\nabla\hat{\mu}$ forms an obtuse angle, a detrimental component emerges in the direction of $\nabla\hat{\mu}$ (indicated by the red dashed arrow). Otherwise, the gradient conflict does not happen, shown as Fig. 2 (b). Consequently, $\lambda$ should be sufficiently large to ensure that the combined force's component in the primary direction remains non-negative, that is

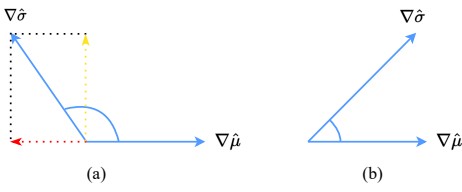

Figure 2: Two situations when updating primary and secondary gradient simultaneously.

$$\lambda\nabla\hat{\mu} + \mathrm{Proj}_{\nabla\hat{\mu}}(\nabla\hat{\sigma}) \geq \epsilon\nabla\hat{\mu} \implies \lambda \geq \epsilon - \frac{\nabla\hat{\mu}\cdot\nabla\hat{\sigma}}{||\nabla\hat{\mu}||^2} := \lambda_1. \tag{6}$$

Here, $\epsilon$ represents the extent to which we wish to update $\nabla\hat{\mu}$ when two gradients are orthogonal. We choose $\epsilon = 1$ in Eq. 6 because it keeps an intact update for the primary gradient $\nabla\hat{\mu}$ in any cases. The harmless component of optimizing $\hat{\sigma}$, illustrated as the dotted yellow arrow in Fig. 2 (a), undergoes with equal strength. The derivation of $\lambda_1$ essentially assumes that the constraint of Eq. 2 is satisfied if $\nabla\hat{\mu} = \mathbf{0}$, which avoids an elaborately specified $\delta$. When $\nabla\hat{\mu} \neq \mathbf{0}$ but $||\nabla\hat{\mu}||$ is small, indicating that the primary objective is nearly minimized, we set $\lambda = \max\{\lambda_1, 0\}$ to prevent negative values.

**Loss view.** Recall that $\hat{\sigma}$ takes $\hat{\mu}$ as input according to Eq. 2, which inspires us to further inspect the combined gradient, denoted by $\nabla$, beyond treating them separately as we do in the gradient view.

**Theorem 2.** *Given the objective of Eq. 2, the combined gradient derived by the update scheme of Eq. 5 can be expressed with an example-reweighting form,*

$$\nabla = \lambda \nabla \hat{\mu} + \nabla \hat{\sigma} = \frac{1}{N} \sum_{i=1}^{N} \underbrace{\left( \lambda + \frac{1}{\hat{\sigma}}(\ell_i - \hat{\mu}) \right)}_{w_i} \frac{\partial \ell_i}{\partial \theta}. \tag{7}$$

The proof of Theorem 2 can be referred to Appendix C. Eq. 7 shows that our fairness formulation with dynamic gradient update implicitly reweights each training example via an unnormalized weight $w_i$, i.e., the Z-score of loss plus a coefficient $\lambda$. This finding connects our work with recent Worst-case fairness studies [13, 12, 22] which up-weight the training examples whose losses are relatively larger, and also a harmless fairness method [20] which directly applies the re-weighting scheme.

As we can see $w_i < 0$ for all examples whose Z-scores below $-\lambda$, raising a concern of unstable optimization [32]. To guarantee that the weight of each training example is non-negative, we require

$$\forall i \in [N] \quad \lambda + \frac{1}{\hat{\sigma}}(\ell_i - \hat{\mu}) \geq 0$$

$$\Longrightarrow \lambda \geq \max_{i \in [N]} \frac{\hat{\mu} - \ell_i}{\hat{\sigma}} = \frac{1}{\hat{\sigma}} \left( \hat{\mu} - \min_{i \in [N]} \ell_i \right) \geq \frac{\hat{\mu}}{\hat{\sigma}} := \lambda_2, \tag{8}$$

where the last inequality holds because predictive losses are typically designed to be non-negative, facilitating the elimination of the sorting procedure.

**Remark 2.** According to Eq. 8, $\lambda_2$ is positive. Notably, $\lambda_2$ can approach 0 if $\hat{\mu} \ll \delta$, where we may obtain a model with good utility but poor fairness. Since Z-scores fall within the range of $-3$ to $+3$ capturing a significant portion (99.7%) of the data in a normal distribution, $\lambda_2$ is often capped by $3$.

In summary, combining Eq. 6 and Eq. 8, we compute the adaptive $\lambda$ at each step that ensures harmless fairness update:

$$\lambda = \max(\lambda_1, \lambda_2) = \max\left( 1 - \frac{\nabla \hat{\mu} \cdot \nabla \hat{\sigma}}{||\nabla \hat{\mu}||^2}, \frac{\hat{\mu}}{\hat{\sigma}} \right). \tag{9}$$

Note that Eq. 9 requires the computation of gradients and values for both $\hat{\mu}$ and $\hat{\sigma}$, which is time-intensive and memory-intensive if executed on the entire dataset. To scale up to large datasets, we provide an efficient mini-batch update strategy. Detailed implementation and algorithm can be referred to Appendix D.

## 4 Experiments

### 4.1 Experimental setup

**Datasets.** Six datasets encompassing binary classification, multi-class classification, and regression are employed. (i) UCI Adult [33], (ii) Law School [34], (iii) COMPAS [35], (iv) CelebA [36], (v) Communities & Crime (C & C) [37], (vi) AgeDB [38]. Note that datasets (i-iii) can be transformed into a logistic regression task by applying MSE loss with the category label as the target. Following the convention established by [24, 15], we select sex (or gender) and race (or Young on CelebA) as sensitive attributes for datasets (i-iv), four attributes for C & C, and one for AgeDB datasets.

**Metrics.** During the evaluation phase, we gain access to the sensitive attributes that partition the dataset into $K$ disjoint groups. As discussed in Section 3.1, our training objective is to uphold a high overall predictive utility level while minimizing group utility disparities to the greatest extent feasible. Henceforth, we assess the performance of our method across five distinct metrics: (i) **Utility**: Overall accuracy for classification (also specified by other metrics like F1-score and prediction error when necessary) and MSE for regression tasks. (ii) **WU**: The worst group utility among all $K$ groups. (iii) **MUD**: Maximum utility disparity, as described in Section 3.1. (iv) **TUD**: Total utility disparity. TUD $= \sum_{k \in [K]} (u_k - \bar{u})$, where $\bar{u}$ is the global average utility. (v) **VAR**: The variance of prediction error. Since we are not able to exhaustively enumerate all possible sensitive attributes and test fairness

via the metrics (ii-iv), VAR necessarily serves as a fairness proxy for any other possible selected sensitive attributes during the test phase. To ensure the reliability of the results, we repeat all the experiments 10 times and average the outcomes.

**Baselines.** We compare VFair against seven baselines including ERM, DRO [13], ARL [24], FairRF [9], MPFR [15], BPF [12], and FKL [39]. Note that DRO, BPF, FairRF, MPFR, and FKL all require some prior demographic information; DRO and BPF necessitate the identification of the worst-off group through a bound of group ratio, while FairRF selects some observed features as pseudo-sensitive attributes, which consequently constrain its application to image datasets (i.e., CelebA); MPFR and FKL which are particularly designed for fair regression tasks also incorporate sensitive attributes. Methods take general loss functions like VFair which apply to both classification tasks and regression tasks, i.e., DRO and ARL are implemented for regression tasks by using the MSE loss. Note that BPF, MPFR, and FKL are not designed with stochastic updates and they suffer from out-of-memory issues under our experimental setup on the UCI Adult and AgeDB datasets. Therefore, the experimental results of this part are not included. Please find more experimental setup details in Appendix E.

Table 1: Comparison of regression results ($\times 10^2$) on five benchmark datasets with the best rank in bold. Here, $\downarrow$ is for Utility and WU because MSE is used, and smaller values indicate better utility.

| | | Utility $\downarrow$ | WU $\downarrow$ | MUD $\downarrow$ | TUD $\downarrow$ | VAR $\downarrow$ |
|---|---|---|---|---|---|---|
| Law School | ERM | $12.88_{(0.12)}$ | $19.75_{(0.21)}$ | $7.33_{(0.14)}$ | $13.45_{(0.21)}$ | $4.89_{(0.07)}$ |
| | DRO | $24.85_{(0.09)}$ | $24.98_{(0.05)}$ | $0.14_{(0.08)}$ | $0.23_{(0.12)}$ | $0$ |
| | ARL | $\mathbf{12.86}_{(0.11)}$ | $19.72_{(0.26)}$ | $7.33_{(0.19)}$ | $13.54_{(0.29)}$ | $4.86_{(0.13)}$ |
| | BPF | $18.75_{(0.50)}$ | $43.25_{(3.44)}$ | $26.33_{(3.20)}$ | $47.33_{(5.16)}$ | $3.98_{(0.29)}$ |
| | MPFR | $13.88$ | $29.39$ | $16.68$ | $32.07$ | $7.15$ |
| | FKL | $13.10$ | $19.37$ | $6.77$ | $13.42$ | $5.01$ |
| | VFair(Ours) | $12.95_{(0.11)}$ | $\mathbf{19.08}_{(0.22)}$ | $\mathbf{6.63}_{(0.18)}$ | $\mathbf{12.53}_{(0.25)}$ | $\mathbf{3.66}_{(0.12)}$ |
| | Improved | -0.07 | +0.67 | +0.7 | +0.92 | +1.23 |
| COMPAS | ERM | $23.08_{(0.67)}$ | $24.49_{(0.76)}$ | $2.50_{(1.17)}$ | $3.45_{(1.76)}$ | $3.23_{(0.8)}$ |
| | DRO | $24.97_{(0.04)}$ | $25.05_{(0.06)}$ | $0.12_{(0.07)}$ | $0.17_{(0.10)}$ | $0$ |
| | ARL | $\mathbf{22.73}_{(0.4)}$ | $24.26_{(0.84)}$ | $2.92_{(1.08)}$ | $3.78_{(1.11)}$ | $3.19_{(0.67)}$ |
| | BPF | $50.80_{(2.18)}$ | $63.46_{(0.99)}$ | $22.87_{(1.69)}$ | $37.77_{(1.84)}$ | $11.18_{(1.11)}$ |
| | MPFR | $36.26$ | $38.36$ | $6.23$ | $9.13$ | $17.33$ |
| | FKL | $28.56$ | $30.49$ | $3.69$ | $6.47$ | $7.58$ |
| | VFair(Ours) | $23.15_{(0.13)}$ | $\mathbf{23.83}_{(0.21)}$ | $\mathbf{0.93}_{(0.21)}$ | $\mathbf{1.17}_{(0.28)}$ | $\mathbf{0.47}_{(0.07)}$ |
| | Improved | -0.07 | +0.66 | +1.57 | +2.28 | +2.76 |
| C & C | ERM | $41.15_{(1.25)}$ | $109.72_{(5.60)}$ | $106.56_{(5.67)}$ | $337.26_{(18.15)}$ | $87.52_{(9.43)}$ |
| | DRO | $99.34_{(3.85)}$ | $257.51_{(40.23)}$ | $248.56_{(48.63)}$ | $715.62_{(189.66)}$ | $284.49_{(72.71)}$ |
| | ARL | $\mathbf{40.43}_{(1.14)}$ | $109.00_{(6.06)}$ | $106.88_{(5.70)}$ | $331.38_{(19.63)}$ | $83.98_{(5.55)}$ |
| | BPF | $71.05_{(1.02)}$ | $127.28_{(4.78)}$ | $110.16_{(10.06)}$ | $320.65_{(26.11)}$ | $96.76_{(8.08)}$ |
| | MPFR | $93.57$ | $296.36$ | $295.59$ | $843.47$ | $375.79$ |
| | FKL | $83.73$ | $278.30$ | $275.29$ | $794.59$ | $321.42$ |
| | VFair(Ours) | $41.17_{(0.64)}$ | $\mathbf{106.40}_{(2.66)}$ | $\mathbf{104.54}_{(3.11)}$ | $\mathbf{318.33}_{(8.96)}$ | $\mathbf{67.44}_{(3.36)}$ |
| | Improved | -0.02 | +3.32 | +2.02 | +18.93 | +20.08 |
| AgeDB | ERM | $4.25_{(0.49)}$ | $4.32_{(0.53)}$ | $0.15_{(0.12)}$ | $0.15_{(0.12)}$ | $0.57_{(0.26)}$ |
| | DRO | $17.72_{(22.59)}$ | $17.98_{(22.65)}$ | $0.5_{(0.37)}$ | $0.5_{(0.37)}$ | $5.76_{(7.67)}$ |
| | ARL | $5.11_{(1.76)}$ | $5.29_{(1.98)}$ | $0.36_{(0.41)}$ | $0.36_{(0.41)}$ | $2.51_{(3.23)}$ |
| | VFair(Ours) | $\mathbf{3.57}_{(0.76)}$ | $\mathbf{3.63}_{(0.77)}$ | $\mathbf{0.12}_{(0.09)}$ | $\mathbf{0.12}_{(0.09)}$ | $\mathbf{0.23}_{(0.08)}$ |
| | Improved | +0.68 | +0.69 | +0.03 | +0.03 | +0.34 |

## 4.2 Examine harmless fairness in regression tasks

Table 1 showcases the comparison results of different methods on regression tasks. The standard deviation calculated from every 10 repeated experiments is presented in the bracket. In the "Improved" row, we computed the improvement of VFair compared to ERM, where "+" denotes improvement rather than a numerical increase. Results with significant changes at the 0.05 significance level are highlighted in green, while others are in yellow. Note that our objective is to gain improvement in fairness metrics while maintaining utility, non-significant changes in utility are desired. However, significant drops in utility violate the harmless setting.

According to Table 1, we have the following findings. (1) VFair significantly improves most fairness metrics with non-significant changes in Utility. Exceptions on C & C and AgeDB are due to their

specific group partition. C & C is split into 16 groups with some extremely small groups, limiting the improvement on WU and MAD while VFair still outperforms others on TAD and VAR. AgeDB is split by gender with a ratio of 4:6, where ERM can also be relatively fair. (2) VFair gains significant VAR improvement on all datasets, guaranteeing that the group utility disparity remains low for any downstream sensitive attributes. (3) The utility of the test set turns out clear distinctions among compared methods because prediction error (MSE) is sensitive to both the possible distribution shift of test data and model parameters. In this sense, only VFair and ARL can still approach the utility of ERM while the rest usually cannot. (4) DRO gains utility close to 0.25 on each group (i.e. a uniform regressor as shown in Fig. 1 (b)) on Law School and COMPAS while using the real prior, shadowed in gray.

### 4.3   Examine harmless fairness in classification tasks

In the context of classification, fairness metrics provide limited improvements without compromising utility (see Appendix F.1 for detailed results). To further investigate the performance gap between regression and classification tasks, we depict classification losses in Fig. 3 following the same scheme in Fig. 1 (b) on real dataset COMPAS. Curves on other datasets are left in Appendix F.2.

**Our observation.** (1) Regardless of the uniform classifier DRO, our method VFair exhibits a more flattened loss curve compared to others while maintaining a comparable area under the curve (filled with pink), signifying a harmless fairness solution. Such results align with our initial idea, as presented in Fig. 1 (b). (2) Our method VFair implies the Worst-case fairness, the average loss of the worst-off group for VFair will be consistently lower than any other method. Our claim is obviously true if the group size is small. Regarding a larger group size, thanks to the fact the total area under each curve is nearly equal and the curve of VFair is always above others at left, we conclude that the worst-off group's losses for VFair are also the lowest. (3) The vertical dotted blue line represents the threshold, where the intersection with the loss curve of VFair values -log(0.5). Divided by it, the samples on

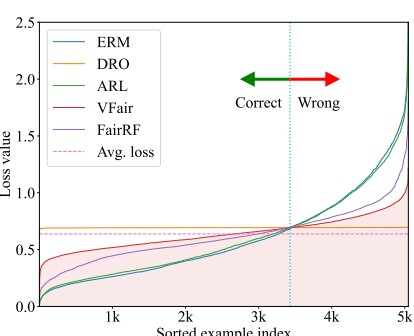

Figure 3: Per-example losses for all compared methods sorted in ascending order on train set.

the left are correctly classified, and conversely on the right. As evidenced by the figure, this threshold is close to each method's intersection. Imagine a situation where the loss curve rotates around the decision point with a smaller angle to the x-axis, obtaining a smaller sample disparity. However, due to the discrete metric and unchanged group partition, the accuracy-based metrics' values for this method remain unchanged after rotation. Therefore, despite our method approaching a horizontal loss curve, thus providing a smaller disparity for any potential group split, the fairness improvement is still bounded by the overall utility.

**Beyond accuracy as utility.** Classifying imbalanced data often applies F1-score as a metric, which is free of the effect on true negative samples which can dominate the accuracy result. We test F1-score performance on UCI Adult and CelebA because they have a remarkable imbalance ratio. The results are summarized in Table 2.

Table 2: Classification results ($\times 100$) comparison on two imbalanced datasets with F1-score as the utility metric. The best rank is highlighted in bold.

|  | UCI Adult | | | | CelebA | | | |
|---|---|---|---|---|---|---|---|---|
|  | Utility↑ | WU ↑ | MUD ↓ | TUD ↓ | Utility↑ | WU ↑ | MUD ↓ | TUD ↓ |
| ERM | 75.02 | 72.17 | 6.87 | 8.88 | 91.40 | 70.17 | 19.39 | 22.82 |
| DRO | 36.27 | 16.06 | 23.59 | 41.17 | 77.52 | 74.29 | **3.9** | **4.78** |
| ARL | 74.90 | 71.85 | 7.32 | 9.49 | 91.60 | 70.39 | 20.14 | 24.33 |
| VFair | **75.98** | **72.74** | **5.82** | **7.40** | **91.91** | **75.70** | 14.39 | 18.50 |

We observe that on UCI Adult, the earned fairness for each fairness method is still limited while on CelebA, VFair yields superior performance. A reasonable explanation is that VFair has the opportunity to discover better solutions in a relatively larger solution space, where more diverse

minima can be examined through fairness criteria. And though F1-score removes the influence of true negative samples, it takes the quantified property and hence may only help amply the gains.

**From quantized to continuous.** For scenarios where prediction error (the difference between prediction and true label) is desired in classification, e.g., assessing whether a model overestimates or underestimates, VFair should be more applicable. To justify this insight, we compare fairness methods (except for FairRF and DRO as they often fall short on utility) on the three datasets reused for regression tasks. Instead of evaluating specific attributes as we do in Section 4.2, we test VFair on all possible divisions of the test set by randomly splitting them into $K$ groups. The three methods are ranked based on their performance under each metric. From the best to the worst, the rank score is 1, 2, and 3. The average rank over 100 times is reported.

Table 3: Average rank of four compared methods. All methods are trained only once.

| | | $K = 4$ | | | | $K = 10$ | | | | $K = 20$ | | | |
| | | Utility | WU | MUD | TUD | Utility | WU | MUD | TUD | Utility | WU | MUD | TUD |
|---|---|---|---|---|---|---|---|---|---|---|---|---|---|
| | ERM | 2.5 | 2.5 | 2.31 | 2.36 | 2.5 | 2.51 | 2.4 | 2.46 | 2.5 | 2.27 | 2.44 | 2.39 |
| UCI Adult | ARL | 2.5 | 2.41 | 2.48 | 2.51 | 2.5 | 2.35 | 2.6 | 2.54 | 2.5 | 2.23 | 2.56 | 2.61 |
| | VFair | **1** | **1.09** | **1.21** | **1.13** | **1** | **1.14** | **1** | **1** | **1** | **1.5** | **1** | **1** |
| | ERM | 2.7 | 2.66 | 2.37 | 2.36 | 2.7 | 2.61 | 2.52 | 2.54 | 2.7 | 2.52 | 2.5 | 2.53 |
| Law School | ARL | **2.3** | 2.34 | 2.36 | 2.37 | 2.3 | 2.39 | 2.48 | 2.46 | 2.3 | 2.31 | 2.5 | 2.47 |
| | VFair | 2.7 | **1** | **1.27** | **1.27** | **1** | **1** | **1** | **1** | **1** | **1.17** | **1** | **1** |
| | ERM | 2.5 | 2.21 | 2.56 | 2.57 | 2.5 | **1.89** | 2.57 | 2.56 | 2.5 | **1.53** | 2.58 | 2.62 |
| COMPAS | ARL | 2.5 | 2.44 | 2.43 | 2.42 | 2.5 | 1.98 | 2.43 | 2.44 | 2.5 | 1.56 | 2.42 | 2.38 |
| | VFair | **1** | **1.35** | **1.01** | **1.01** | **1** | 2.13 | **1** | **1** | **1** | 2.91 | **1** | **1** |

Results in Table 3 show that our method VFair has a better rank than other methods regardless of the choice of $K$, demonstrating that VFair prefers the utility metrics that are loss/error-related. As mentioned in Section 4.1, VAR serves as an approximation for an extreme group split, where each group consists of only one member. Thus, the significantly low VAR in Table 1 and Table 5 implies good results in random partitions on the test set, evidencing that variance can serve as an effective optimized term in Rawlsian fairness tasks without prior demographic information.

## 4.4 A closer look at VFair

We examine our VFair through extensive experiments. Here we present the partial results and main conclusions. One can refer to Appendix F.2, F.3, and F.4 for more details.

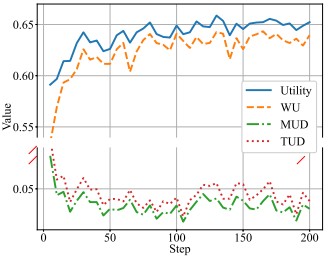
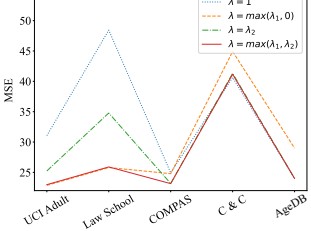
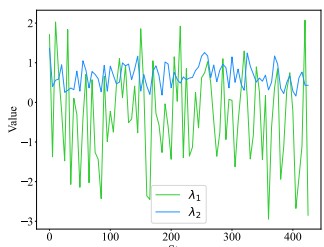

(a) Training results on COMPAS in the classification task.

(b) MSE on five regression datasets. Lower is better.

(c) The curves of $\lambda_1$ and $\lambda_2$ during training on C & C.

Figure 4: Experimental verification of the harmless update strategy.

**Method efficacy.** We monitor the performance of VFair during the training phase by evaluating it with four utility-related metrics on the test set of COMPAS. Fig. 4 (a) indicates these curves naturally improve in the desired direction under the variance penalty, verifying the effectiveness of our method.

**Ablation study.** We train our model under four settings: $\lambda = 1$, $\lambda = \max(\lambda_1, 0)$, $\lambda = \lambda_2$, and $\lambda = \max(\lambda_1, \lambda_2)$. As depicted in Fig. 4 (b), we present the per-dataset utility on five regression datasets (results are proportionally scaled on each dataset for a clearer presentation). The full version, considering both $\lambda_1$ and $\lambda_2$, exhibits the most stability in preserving low MSE, enabling a harmless

solution. In Fig. 4 (c), we demonstrate an example of $\lambda_1$ and $\lambda_2$ on C & C dataset during training, where both serve distinct and complementary roles in preventing the model from sacrificing utility.

**Model examination.** We scrutinize the fair models by studying their parameters and prediction similarity with ERM. Our experiments found that the model learned by VFair is more dissimilar from ERM than other methods. For example, on Law School, the cosine similarity of model parameters in ARL and VFair with ERM is 0.6106 and 0.5839, respectively. This indicates that VFair may explore a larger model space to achieve better performance.

## 5   Conclusion

Towards harmless Rawlsian fairness regardless of demographics, we have introduced a straightforward yet effective variance-based method VFair. VFair harnesses the principle of decreasing the variance of losses to steer the model's learning trajectory, thereby bridging the utility gaps appearing at potential group partitions. The optimization with a devised dynamic weight parameter operated at both the loss and gradient levels, ensuring the model converges at the fairest point within the optimal solution set. By capping the Z-score, our dynamic weight parameter can also prevent the model from overfocusing on outliers with larger losses. The experiments affirm that regression can be a prior-free task to Rawlsian harmless fairness because error-based metrics are more consistent with loss. Strong prior for demographics may be needed for quantized metrics like accuracy in classification tasks. As discussed in Appendix G, limitations may arise from computational costs, where VFair takes twice the time of ERM to uncover more information without access to demographic prior. Future work will involve identifying and addressing further challenges that may arise when applying VFair for the prediction of non-IID data.

## Acknowledgement

This research is supported by the National Research Foundation, Singapore and Infocomm Media Development Authority under its Trust Tech Funding Initiative (No. DTC-RGC-04). Any opinions, findings and conclusions or recommendations expressed in this material are those of the author(s) and do not reflect the views of National Research Foundation, Singapore and Infocomm Media Development Authority.

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

# A Proof of Proposition 1

**Proposition 1.** $u \perp s$ *holds for any $s$ that splits data into a number of groups, if and only if the loss $\ell$ is (approximately) independent of the training example $z$, i.e., $\ell \perp z$.*

*Proof.* Suppose that $s$ splits data into $K$ disjoint groups. Let the random variable $k$ represent the group index. We can rephrase the statement as $\forall s, u \perp k|s \Leftrightarrow \ell \perp z$, which is established through the following two steps.

Step 1. Since proving "$\forall s, u \perp k|s \Rightarrow \ell \perp z$" is difficult, we consider its contrapositive, i.e., "$\ell \not\perp z \Rightarrow \exists s, u \not\perp k$". If the value of $\ell$ spreads across a large range, indicating some examples are well-fitted (small loss) while others are not (large losses), we can simply let $s$ split them according to if well-fitted. Since $u_1 \neq u_2$, $u \not\perp k$ follows.

Step 2. The assertion, "$\ell \perp z \Rightarrow \forall s, u \perp k$", is true when the condition $\ell \perp z$ is strictly satisfied. Particularly, if a quantized utility is applied, e.g., accuracy, the assertion holds even if we relax the condition–$\ell$ exhibits approximate dependence on $z$. Two distinct scenarios arise. (i) All losses are concentrated in proximity to the decision boundary, resembling the characteristics of a uniform classifier. In the context of a finite partition by $s$, the accuracy of each subgroup within a uniform classifier statistically converges towards $0.5$ for a binary classification. (ii) All losses are conspicuously distanced from the decision boundary, akin to an ideal classifier. In this case, an ideal classifier consistently achieves a subgroup accuracy of $1$, irrespective of the chosen split. In both situations, we can indeed conclude that $\forall s, u \perp k|s$. $\qquad\square$

# B Fairness objective

## B.1 Proof of Theorem 1

To prove Theorem 1, we need the following lemma.

**Lemma 1.** *Given $N$ non-negative numbers $\{v_i\}_{i=1}^N$, the following inequality holds*

$$\max_{i \in [N]} v_i - \min_{i \in [N]} v_i \leq \sum_{i=1}^{N-1} |v_i - v_{i+1}|. \tag{10}$$

*Proof.* Let $a = \min\left(\arg\min_{i \in [N]} v_i, \arg\max_{i \in [N]} v_i\right)$, $b = \max\left(\arg\min_{i \in [N]} v_i, \arg\max_{i \in [N]} v_i\right)$. According to the triangle inequality, we have

$$\max_{i \in [N]} v_i - \min_{i \in [N]} v_i = |\max_{i \in [N]} v_i - \min_{i \in [N]} v_i| \leq \sum_{i=a}^{b} |v_i - v_{i+1}|$$
$$\leq \sum_{i=1}^{N-1} |v_i - v_{i+1}|.$$

Particularly, the equality holds if $\{v_i\}_{i=1}^N$ are arranged in monotonic order. $\qquad\square$

**Theorem 1.** $\forall s \in \mathcal{S}, \forall \theta \in \Theta, \ell_{MUD} \leq C\sqrt{\mathbb{V}_z[\ell(z;\theta)]}$, *where $C$ is a constant.*

*Proof.* Let $r_k$ denote the expected loss of $k$-th group. We have

$$\ell_{\text{MUD}} := \max_{k \in [K]} r_k - \min_{k \in [K]} r_k \leq \max_{i \in [N]} \ell_i - \min_{i \in [N]} \ell_i$$

$$\stackrel{①}{\leq} \sum_{i=1}^{N-1} |\ell_i - \ell_{i+1}| \tag{11}$$

$$\leq \sum_{i<j}^{N} |\ell_i - \ell_j| \stackrel{②}{\leq} \sqrt{C_N^2 \sum_{i<j}^{N} |\ell_i - \ell_j|^2}$$

$$\stackrel{③}{=} N\sqrt{C_N^2 \mathbb{V}_z[\ell(z;\theta)]} \tag{12}$$

$$\stackrel{④}{\leq} \frac{C_N^2 N^2}{L} \mathbb{V}_z[\ell(z;\theta)] \tag{13}$$

where ① follows inequality given by Lemma 1, ② uses the norm inequality of $||x||_1 \leq \sqrt{dim(x)}||x||_2$, ③ is derived from the fact that $\mathbb{V}_z[\ell(z;\theta)] = \frac{1}{N^2}\sum_{i<j}^{N}(\ell_i - \ell_j)^2$, and ④ is similar to ② and further uses $\sum_{i<j}^{N} |\ell_i - \ell_j| \geq L$. □

Note that Theorem 1 adopts the result of Eq. 12 which scales one side by a factor $N$, making it not a very tight bound. However, we justify that this option is more efficient than others in the next subsection. Additionally, although we start with $\ell_{\text{MUD}}$, it is easy to verify that the derived bound also serves as a proxy for other utility disparity metrics, e.g., TUD.

### B.2 Option of loss for Rawlsian fairness

To compare the efficacy of three forms of fairness loss, i.e., Eqs. 11, 12, and 13, we denote each as $\hat{\pi}$, $\hat{\sigma}^2$, and $\hat{\sigma}$ respectively:

- $\hat{\pi} = \sum_{i=1}^{N-1} |\ell_i - \ell_{i+1}|$ *(Pairwise)*

- $\hat{\sigma} = \frac{1}{\sqrt{N}}\sqrt{\sum_{i=1}^{N}(\ell_i - \hat{\mu})^2}$ *(Standard deviation)*

- $\hat{\sigma}^2 = \frac{1}{N}\sum_{i=1}^{N}(\ell_i - \hat{\mu})^2$ *(Variance)*

Here, we analyze the choice of the objective from both theoretical and experimental levels. The experiment results are shown in Table 4.

Table 4: Comparison of three fairness objectives on four benchmark datasets, where the utility-based results are with %, and the results of VAR are $\times 10^2$ for a neat presentation. Best results are in bold.

| | Objective | Utility ↑ | WU↑ | MUD↓ | TUD↓ | VAR↓ |
|---|---|---|---|---|---|---|
| | $\hat{\pi}$ | 82.98 | 78.35 | 16.19 | 21.10 | **0** |
| UCI Adult | $\hat{\sigma}^2$ | 84.70 | 80.34 | 15.72 | 20.79 | 7.18 |
| | $\hat{\sigma}$ | **84.74** | **80.36** | **15.71** | **20.71** | 8.17 |
| | $\hat{\pi}$ | 84.05 | 72.96 | 11.92 | 22.51 | **0.03** |
| Law School | $\hat{\sigma}^2$ | 85.33 | 74.60 | 11.67 | 20.91 | 6.91 |
| | $\hat{\sigma}$ | **85.40** | **74.81** | **11.24** | **20.31** | 19.35 |
| | $\hat{\pi}$ | 55.78 | 51.60 | 8.70 | 12.24 | **0** |
| COMPAS | $\hat{\sigma}^2$ | 63.45 | 59.14 | 8.71 | 11.36 | 0.04 |
| | $\hat{\sigma}$ | **66.80** | **63.86** | **6.25** | **8.47** | 1.86 |
| | $\hat{\pi}$ | 44.88 | 20.16 | 49.16 | 53.85 | **0** |
| CelebA | $\hat{\sigma}^2$ | 92.45 | 89.53 | 3.44 | 4.63 | 14.4 |
| | $\hat{\sigma}$ | **93.43** | **91.09** | **2.73** | **3.85** | 11.7 |

**Pairwise difference objective.** Eq. 11 computes the consecutive pairwise difference of all training losses. Putting it under the proposed harmless update, we need to particularly get the dynamic value for $\lambda_2$.

Similar to Eq. 8, the $\lambda_2$ is set to guarantee samples will not be assigned negative weights.

$$\nabla \hat{\pi} = \nabla \sum_{i=1}^{N-1} |\ell_i - \ell_{i+1}| = \nabla \sum_{i=1}^{N} \phi_i \ell_i = \sum_{i=1}^{N} \phi_i \frac{\partial \ell_i}{\partial \theta} \tag{14}$$

By inspecting the possible combination of the absolute equation, it is obvious that $\phi_1, \phi_N \in \{-1, 1\}$, and $\phi_i \in \{-2, 0, 2\}, \forall i \in (1, N)$. Consequently, $\lambda_2$ can be calculated as:

$$\forall i \in [N] \quad \lambda + \phi_i \geq 0 \implies \lambda \geq 2 := \lambda_2 \tag{15}$$

When operated on a mini-batch, unlike $\hat{\sigma}$ and $\hat{\sigma}^2$, the pairwise difference objective does not consider global information, losing the relative relationships when attempting to identify the Worst-case group (also discussed in Appendix D). The instability arising from the difference of pairwise sample losses might mislead the upgrading process, as evidenced in our experiments. On challenging datasets such as COMPAS and CelebA, the model tends to converge towards a uniform classifier, even constrained by dynamic parameters.

**Variance objective.** By dropping constant factor of Eq. 13, we employ $\hat{\sigma}^2 = \frac{1}{N} \sum_{i=1}^{N} (\ell_i - \hat{\mu})^2$ as secondary objective. Similar to the proof of Theorem 2, we have

$$\begin{aligned} \nabla &= \lambda \nabla \hat{\mu} + \nabla \hat{\sigma}^2 \\ &= \frac{1}{N} \sum_{i=1}^{N} \left( \lambda + 2\left(\ell_i - \hat{\mu}\right) \right) \frac{\partial \ell_i}{\partial \theta} \end{aligned} \tag{16}$$

Consequently, we get $\lambda_2 = 2\left(\hat{\mu} - \min_{i \in [N]} \ell_i\right)$.

It can be observed from Eq. 13 that $\hat{\sigma}^2$ serves as a broader constraint for $\ell_{\text{MUD}}$. As a result, it is a less restrictive objective for group disparity compared to $\hat{\sigma}$. However, the square-version term penalizes more on both smaller and larger losses, resulting in an unavoidable decrease in overall utility (e.g., unreliable data with spurious correlation) and hence on all utility-based fairness metrics. Please see the evidential experiments in Table 4 that using $\hat{\sigma}^2$ as objective results in lower variance but higher group disparity.

## C Proof of Theorem 2

**Theorem 2.** *Given the objective of Eq. 2, the combined gradient derived by the update scheme of Eq. 5 can be expressed with an example-reweighting form,*

$$\nabla = \lambda \nabla \hat{\mu} + \nabla \hat{\sigma} = \frac{1}{N} \sum_{i=1}^{N} \underbrace{\left( \lambda + \frac{1}{\hat{\sigma}} (\ell_i - \hat{\mu}) \right)}_{w_i} \frac{\partial \ell_i}{\partial \theta}.$$

*Proof.* Based on the form of $\hat{\mu}(\theta)$ and $\hat{\sigma}(\theta)$ in Eq. 2, we have

$$\nabla \hat{\mu} = \frac{1}{N} \sum_{i=1}^{N} \frac{\partial \ell_i}{\partial \theta} \tag{17}$$

$$\nabla\hat{\sigma} = \frac{1}{2\sqrt{\frac{1}{N}\sum_{i=1}^{N}(\ell_i - \hat{\mu})^2}} \frac{1}{N}\sum_{j=1}^{N} 2(\ell_j - \hat{\mu})\frac{\partial(\ell_j - \hat{\mu})}{\partial\theta}$$

$$= \frac{1}{N}\sum_{j=1}^{N} \frac{\ell_j - \hat{\mu}}{\hat{\sigma}} \frac{\partial\ell_j}{\partial\theta} - \frac{1}{N\hat{\sigma}}\sum_{j=1}^{N}\left((\ell_j - \hat{\mu})\frac{1}{N}\sum_{k=1}^{N}\frac{\partial\ell_k}{\partial\theta}\right)$$

$$= \frac{1}{N}\sum_{j=1}^{N}\frac{\ell_j - \hat{\mu}}{\hat{\sigma}}\frac{\partial\ell_j}{\partial\theta} \tag{18}$$

$$- \frac{1}{N\hat{\sigma}}\underbrace{\left(\left(\sum_{j=1}^{N}\ell_j\right) - N\hat{\mu}\right)}_{=0}\left(\frac{1}{N}\sum_{k=1}^{N}\frac{\partial\ell_k}{\partial\theta}\right)$$

The example-reweighting form of gradients follows by unifying Eq. 17 and 18. □

## D  Implementation and algorithm

To enable the application on large datasets, we provide a mini-batch update strategy. It is worth noting that the mean loss encompasses global information that could guide the update direction for each sample. The variance on a mini-batch computed on a local mean loss may cause unstable optimization, especially when the batch size is small. As such, we consider maintaining a global mean which assists with the mini-batch update. To this end, we employ Exponential Moving Average (EMA) as an approximation for the global mean loss:

$$\hat{\mu}^t = \beta\hat{\mu}^{t-1} + \frac{1-\beta}{b}\sum_{i=1}^{b}\ell_i, \tag{19}$$

where the decay parameter $\beta$ is set 0.99 for all datasets in our experiments, and $b$ denotes batch size. The comprehensive implementation of our VFair is elucidated in the following algorithm.

---

**Algorithm 1** Harmless Rawlsian Fairness without Demographics via VFair.

---

**Input:** Training set $\mathcal{D} = \{z_i\}_{i=1}^{N}$, where $z_i = (x_i, y_i) \in \mathcal{X} \times \mathcal{Y}$
**Output:** Learned model parameterized by $\theta \in \Theta$
 1: Initialize parameters $\theta$
 2: Initialize $\hat{\mu}^0 \leftarrow 0$
 3: **for** epoch $\leftarrow 1$ **to** $N_{\text{epochs}}$ **do**
 4:    **for** mini-batch $\mathcal{B} \subset \mathcal{D}$ **do**
 5:       Compute the losses $\{\ell_i\}_{i=1}^{b}$
 6:       Update $\hat{\mu}^t$ as in Eq. 19
 7:       Update $\hat{\sigma} \leftarrow \sqrt{\frac{1}{b}\sum_{i=1}^{b}(\ell_i - \hat{\mu}^t)^2}$
 8:       Compute primary gradient $\nabla\hat{\mu}$
 9:       Compute secondary gradient $\nabla\hat{\sigma}$
10:       Compute $\lambda_1$ as in Eq. 6
11:       Compute $\lambda_2$ as in Eq. 8
12:       Compute dynamic $\lambda^t$ as in Eq. 9
13:       Update parameters $\theta$ as in Eq. 5
14:    **end for**
15: **end for**

---

## E  Experimental setup details

All the deep-learning-based models, excluding FairRF, which operates within a distinct problem setting, conform to a shared neural network framework. Specifically, for binary classification tasks, the core neural network architecture consists of an embedding layer followed by two hidden layers,

with 64 and 32 neurons, respectively. In the ARL model, an additional adversarial component is integrated, detailed in its respective paper, featuring one hidden layer with 32 neurons. For multi-classification tasks, the primary neural network transforms into Resnet18, and the embedding layer transitions to a Conv2d-based frontend. Throughout these experiments, the Adagrad optimizer was employed. FairRF, utilizing its officially published code implementation, maintains the same backbone network with nuanced variations in specific details. Particularly, fair regression methods MPFR and FKL are implemented adapted from [15].

As For the loss function, we implemented Binary Cross-Entropy, Cross-Entropy, and Mean Square Error for binary classification, multi-class classification, and regression tasks, respectively. Note that our method is general and can be compatible with any other forms of loss.

All experiments were conducted on Ubuntu 20.04 with one NVIDIA GeForce RTX 3090 graphics processing unit (GPU), which has a memory capacity of 24 GB.

To compare all baselines under the harmless fairness setting, we implement them into the same scheme and select the epoch with the nearest loss compared to a converged ERM. Detailedly, each method has an empirical loss, which in our method is denoted as $\hat{\mu}$ and in ARL is denoted as learner loss (compared to adversarial loss). Based on this loss, we select the harmless epoch which has the nearest loss value compared to a well-trained ERM model.

# F More experimental results

## F.1 Experimental results on classification tasks

Table 5: Comparison of classification results on four benchmark datasets, where the results of utility (i.e., accuracy) based metrics are with % and the results of VAR are $\times 10^2$ for a neat presentation.

| | | Utility ↑ | WU ↑ | MUD ↓ | TUD ↓ | VAR ↓ |
|---|---|---|---|---|---|---|
| UCI Adult | ERM | $84.67_{(0.58)}$ | $80.20_{(0.82)}$ | $16.13_{(0.82)}$ | $20.78_{(0.99)}$ | $33.89_{(4.77)}$ |
| | DRO | $74.39_{(9.74)}$ | $69.82_{(0.36)}$ | $16.89_{(0.35)}$ | $27.26_{(0.24)}$ | **0** |
| | ARL | $84.60_{(0.63)}$ | $80.11_{(0.91)}$ | $16.17_{(1.05)}$ | $20.91_{(0.95)}$ | $36.18_{(8.41)}$ |
| | FairRF | $84.27_{(0.13)}$ | $80.01_{(0.15)}$ | $15.73_{(0.18)}$ | $\mathbf{20.26}_{(0.58)}$ | $25.83_{(1.38)}$ |
| | VFair | $\mathbf{84.74}_{(0.34)}$ | $\mathbf{80.36}_{(0.49)}$ | $\mathbf{15.71}_{(0.73)}$ | $20.71_{(0.80)}$ | $08.17_{(0.98)}$ |
| | p-value | 0.08 | 0.54 | 0.24 | 0.86 | 0 |
| Law School | ERM | $\mathbf{85.59}_{(0.67)}$ | $74.49_{(1.84)}$ | $12.08_{(2.74)}$ | $21.50_{(3.35)}$ | $36.95_{(1.37)}$ |
| | DRO | $59.76_{(9.69)}$ | $52.28_{(5.07)}$ | $\mathbf{10.49}_{(6.31)}$ | $\mathbf{16.56}_{(11.87)}$ | 244.81 |
| | ARL | $85.27_{(0.71)}$ | $74.78_{(2.12)}$ | $11.52_{(2.21)}$ | $21.52_{(1.97)}$ | $37.95_{(1.80)}$ |
| | FairRF | $81.91_{(0.27)}$ | $68.75_{(1.61)}$ | $14.48_{(1.65)}$ | $26.84_{(2.20)}$ | $30.80_{(1.59)}$ |
| | VFair | $85.40_{(0.99)}$ | $\mathbf{75.25}_{(1.51)}$ | $11.00_{(1.92)}$ | $19.91_{(2.43)}$ | $\mathbf{06.29}_{(0.24)}$ |
| | p-value | 0.62 | 0.33 | 0.32 | 0.24 | 0 |
| COMPAS | ERM | $66.70_{(0.66)}$ | $63.20_{(1.64)}$ | $07.15_{(1.46)}$ | $09.12_{(1.79)}$ | $15.63_{(3.38)}$ |
| | DRO | $24.97_{(0.50)}$ | $25.05_{(1.27)}$ | $0.12_{(1.08)}$ | $0.17_{(1.77)}$ | 0 |
| | ARL | $66.65_{(0.55)}$ | $63.27_{(1.99)}$ | $06.93_{(1.83)}$ | $09.09_{(3.71)}$ | $14.42_{(3.64)}$ |
| | FairRF | $62.90_{(0.43)}$ | $61.55_{(1.06)}$ | $\mathbf{02.64}_{(1.55)}$ | $\mathbf{03.69}_{(2.1)}$ | $06.93_{(1.26)}$ |
| | VFair | $\mathbf{66.80}_{(0.27)}$ | $\mathbf{63.86}_{(0.57)}$ | $06.25_{(0.8)}$ | $08.47_{(1.23)}$ | $\mathbf{1.86}_{(0.12)}$ |
| | p-value | 0.66 | 0.24 | 0.1 | 0.36 | 0 |
| CelebA | ERM | 92.80 | 89.77 | 03.64 | 04.77 | 40.08 |
| | DRO | 83.97 | 82.19 | **2.37** | **2.7** | 21.48 |
| | ARL | 93.26 | 89.84 | 04.02 | 05.41 | 37.38 |
| | FairRF | - | - | - | - | - |
| | VFair | **93.43** | **91.09** | 02.74 | 03.85 | **11.70** |

From the experimental results in Table 5, we can observe that: (1) VFair, without any prior, consistently achieves top-2 performances in classification tasks, competing with or outperforming baselines that use priors, e.g., DRO and FairRF. However, except for VAR, metrics earn limited improvements.

We calculated the p-value for each metric between ERM and VFair to quantify the limitation. The results show that the p-value of metrics, except for VAR, remains high. Generally, a p-value less than 0.05 is considered indicative of a significant difference between the two groups. Even with the same datasets, especially on COMPAS, it is found that harmless Rawlsian fairness is difficult to earn for classification while comparatively easier for regression tasks. (2) From the Utility dimension, FairRF and DRO sometimes fail to guarantee a comparable utility, because constraining group fairness on their proxy attributes unavoidably hurts the overall model performance. With this cost, they sometimes achieve a noteworthy fairness improvement. Note that DRO turns into a uniform classifier on COMPAS, shadowed in gray. (3) CelebA seems an exception where VFair attains meaningful fairness improvement while others do not. A reasonable explanation is that VFair has the opportunity to discover better solutions in a relatively larger solution space, where more diverse minima can be examined through fairness criteria. (4) We also notice that because we explicitly optimize variance, VAR has been remarkably decreased in VFair across all datasets, showing flattened prediction errors on all test sets.

## F.2 VFair training curves

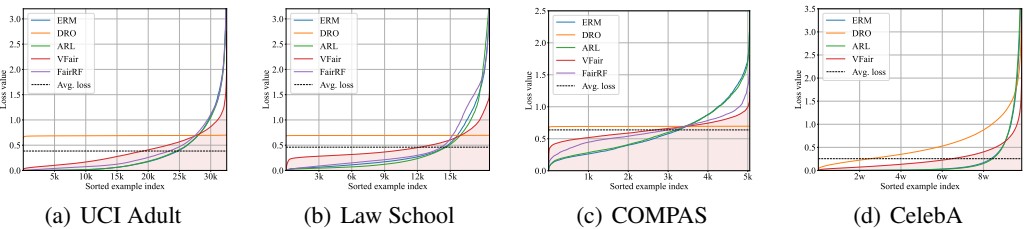

(a) UCI Adult  (b) Law School  (c) COMPAS  (d) CelebA

Figure 5: Full version of per-example losses for all compared methods sorted in ascending order on the training set of four benchmark classification datasets. Dash lines represent their average losses.

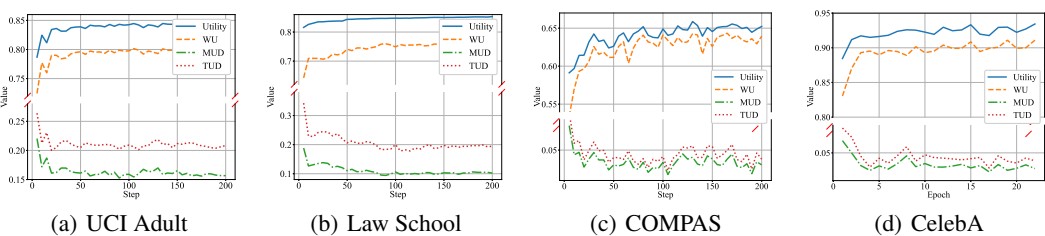

(a) UCI Adult  (b) Law School  (c) COMPAS  (d) CelebA

Figure 6: Full version of test performance curves of four utility-based fairness metrics during the training process (Step/5) on four benchmark datasets with Accuracy served as the utility.

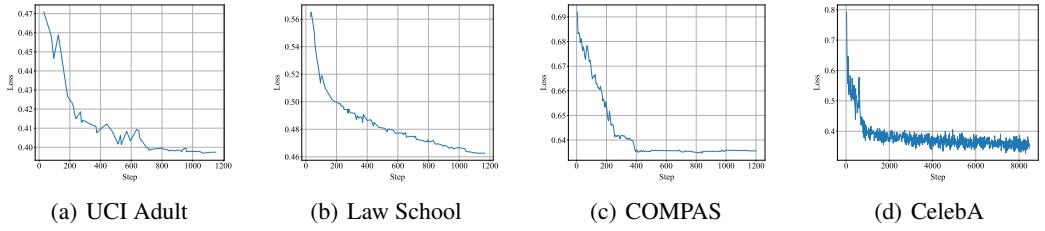

(a) UCI Adult  (b) Law School  (c) COMPAS  (d) CelebA

Figure 7: The loss curve of primary objective during the training process on four benchmark datasets.

We depict the full version of the per-sample losses for all compared methods sorted in ascending order on the training set in Fig. 5. From Fig. 5, we surprisingly see that across different datasets our VFair is the unique one that has the flattened curve while DRO, ARL, and FairRF are essentially close to ERM.

The full version of test performance curves of four utility-related metrics during the training process on four benchmark datasets are present in Fig. 6. Our VFair effectively improves all utility-related metrics.

Fig. 7 illustrates the convergence of training loss on four benchmark datasets. As the final combined objective is updated directly at the gradient level, which does not have a unified loss form, we display the curve of losses of the primary objective, representing the model's utility. We observe that our upgrading method in Eq. 7 effectively steers the model towards convergence. Note that our dynamic updating strategy is similar to [28], which is theoretically proven to converge.

## F.3 Detailed ablation results

According to the ablation setting in Section 4.4, we conducted throughout experiments on the classification task and the regression task, respectively.

As shown in Table 6, our method already achieves competitive results by solely employing $\lambda_2$. The full version which integrates both $\lambda_1$ and $\lambda_2$ demonstrates more stable results. Notably, on the Law School and COMPAS datasets, there exist situations when the model converges towards a uniform classifier, as indicated by the gray region. These uniform classifiers predict all the samples near the decision boundary, causing their losses to share very similar values and form variances at a scale of around $1e-7$. This phenomenon underscores the effectiveness of $\lambda_2$ in preventing the model from collapsing to a low-utility model. Moreover, by adding $\lambda_1$, our method consistently improved in four utility-related metrics. These results show that $\lambda_1$ effectively guides the model to converge to a better point at the gradient level.

Table 6: Comparison of classification ablation results (%) on four benchmark datasets. All of the results are averaged over 10 repeated experiments to mitigate randomness, with the best results highlighted in red and the second-best in blue (excluding the uniform situation).

| | $\lambda_1$ | $\lambda_2$ | Utility ↑ | WU ↑ | MUD ↓ | TUD ↓ | VAR ↓ |
|---|---|---|---|---|---|---|---|
| **UCI Adult** | $\lambda=1$ | | $84.71_{(0.32)}$ | $80.36_{(0.44)}$ | $15.85_{(0.65)}$ | $20.94_{(0.72)}$ | $3_{(0.32)}$ |
| | | ✓ | $84.68_{(0.36)}$ | $80.27_{(0.50)}$ | $15.91_{(0.63)}$ | $20.97_{(0.89)}$ | $7.75_{(0.81)}$ |
| | ✓ | | $84.52_{(0.44)}$ | $80.08_{(0.65)}$ | $15.99_{(0.73)}$ | $20.92_{(0.56)}$ | $6.58_{(1.15)}$ |
| | ✓ | ✓ | $84.74_{(0.34)}$ | $80.36_{(0.49)}$ | $15.71_{(0.73)}$ | $20.71_{(0.80)}$ | $8.17_{(0.98)}$ |
| **Law School** | $\lambda=1$ | | $84.36_{(0.11)}$ | $74.30_{(0.84)}$ | $10.88_{(0.95)}$ | $20.73_{(1.63)}$ | $0.05_{(0.02)}$ |
| | | ✓ | $85.40_{(0.30)}$ | $75.09_{(0.58)}$ | $11.20_{(0.82)}$ | $20.43_{(1.66)}$ | $6.3_{(0.14)}$ |
| | ✓ | | $45.39_{(28.53)}$ | $32.03_{(14.79)}$ | $30.31_{(3.41)}$ | $53.12_{(5.08)}$ | $0_{(0)}$ |
| | ✓ | ✓ | $85.40_{(0.99)}$ | $75.25_{(1.51)}$ | $11.00_{(1.92)}$ | $19.91_{(2.43)}$ | $6.29_{(0.24)}$ |
| **COMPAS** | $\lambda=1$ | | $55.21_{(2.43)}$ | $49.90_{(3.44)}$ | $10.51_{(4.34)}$ | $13.28_{(5.51)}$ | $0_{(0)}$ |
| | | ✓ | $64.29_{(0.99)}$ | $60.44_{(3.63)}$ | $7.34_{(3.76)}$ | $9.67_{(4.87)}$ | $0.03_{(0.02)}$ |
| | ✓ | | $66.45_{(0.85)}$ | $63.49_{(1.90)}$ | $6.60_{(2.40)}$ | $8.40_{(3.12)}$ | $1.91_{(0.24)}$ |
| | ✓ | ✓ | $66.80_{(0.27)}$ | $63.86_{(0.57)}$ | $6.25_{(0.80)}$ | $8.47_{(1.23)}$ | $1.86_{(0.12)}$ |
| **CelebA** | $\lambda=1$ | | $92.04$ | $89.22$ | $3.66$ | $4.65$ | $0.1269$ |
| | | ✓ | $93.46$ | $90.62$ | $3.49$ | $4.67$ | $0.1161$ |
| | ✓ | | $93.23$ | $90.08$ | $3.70$ | $5.14$ | $0.0753$ |
| | ✓ | ✓ | $93.43$ | $91.09$ | $2.73$ | $3.85$ | $0.1170$ |

As shown in Table 7, simply employing $\lambda=1$ or $\lambda_1$ can reach more significant fairness improvement, but at the cost of sacrificing utility. The full version considering both $\lambda_1$ and $\lambda_2$ is the only one containing harmless on all datasets, which was discussed in Section 4.4.

## F.4 Model similarity with ERM.

We examine the similarity between fair models and an ERM model. We conduct experiments comparing ERM, DRO, ARL (without adversary network), and VFair, as they share the same model structure. By calculating the Cosine similarity of model parameters and prediction similarity with ERM as a reference, we get results shown in Table 8.

Table 7: Comparison of regression ablation results (%) on four benchmark datasets. All of the results are averaged over 10 repeated experiments to mitigate randomness, with the best results highlighted in red and the second-best in blue (excluding the uniform situation).

| | $\lambda_1$ | $\lambda_2$ | Utility ↓ | WU ↓ | MUD ↓ | TUD ↓ | VAR ↓ |
|---|---|---|---|---|---|---|---|
| UCI Adult | $\lambda=1$ | | $15.53_{(0.64)}$ | $17.47_{(0.46)}$ | $6.94_{(0.71)}$ | $11.27_{(1.22)}$ | $0.89_{(0.18)}$ |
| | | ✓ | $12.62_{(0.28)}$ | $15.26_{(0.18)}$ | $9.55_{(0.53)}$ | $15.50_{(0.88)}$ | $1.63_{(0.22)}$ |
| | ✓ | | $11.43_{(0.25)}$ | $14.14_{(0.25)}$ | $9.78_{(0.44)}$ | $15.97_{(0.75)}$ | $2.44_{(0.40)}$ |
| | ✓ | ✓ | $11.49_{(0.27)}$ | $14.24_{(0.34)}$ | $9.92_{(0.39)}$ | $16.25_{(0.73)}$ | $2.52_{(0.43)}$ |
| Law School | $\lambda=1$ | | $24.22_{(0.13)}$ | $24.55_{(0.07)}$ | $0.36_{(0.07)}$ | $0.66_{(0.13)}$ | $0.01_{(0)}$ |
| | | ✓ | $17.39_{(0.31)}$ | $20.77_{(0.29)}$ | $3.73_{(0.08)}$ | $7.10_{(0.20)}$ | $0.85_{(0.07)}$ |
| | ✓ | | $12.90_{(0.12)}$ | $19.01_{(0.15)}$ | $6.60_{(0.13)}$ | $12.51_{(0.21)}$ | $3.63_{(0.13)}$ |
| | ✓ | ✓ | $12.95_{(0.11)}$ | $19.08_{(0.22)}$ | $6.63_{(0.18)}$ | $12.53_{(0.25)}$ | $3.66_{(0.12)}$ |
| COMPAS | $\lambda=1$ | | $24.92_{(0.06)}$ | $25.03_{(0.11)}$ | $0.24_{(0.13)}$ | $0.34_{(0.17)}$ | $0.02_{(0.01)}$ |
| | | ✓ | $23.17_{(0.16)}$ | $23.86_{(0.20)}$ | $0.91_{(0.20)}$ | $1.18_{(0.28)}$ | $0.45_{(0.07)}$ |
| | ✓ | | $24.79_{(0.04)}$ | $24.92_{(0.10)}$ | $0.24_{(0.11)}$ | $0.31_{(0.13)}$ | $0.02_{(0.01)}$ |
| | ✓ | ✓ | $23.15_{(0.13)}$ | $23.83_{(0.21)}$ | $0.93_{(0.21)}$ | $1.17_{(0.28)}$ | $0.47_{(0.07)}$ |
| C & C | $\lambda=1$ | | $40.63_{(0.67)}$ | $106.52_{(1.92)}$ | $105.58_{(2.06)}$ | $315.92_{(7.62)}$ | $68.69_{(3.75)}$ |
| | | ✓ | $41.32_{(0.65)}$ | $106.28_{(2.26)}$ | $104.22_{(2.85)}$ | $315.11_{(7.19)}$ | $66.96_{(3.05)}$ |
| | ✓ | | $44.92_{(1.21)}$ | $106.21_{(2.65)}$ | $96.55_{(2.11)}$ | $299.77_{(5.95)}$ | $60.54_{(2.37)}$ |
| | ✓ | ✓ | $41.17_{(0.64)}$ | $106.40_{(2.66)}$ | $104.54_{(3.11)}$ | $318.33_{(8.96)}$ | $67.44_{(3.36)}$ |

Table 8: Similarity between fair models and an ERM model on three benchmark datasets.

| | Cosine Similarity | | | Prediction Similarity | | |
|---|---|---|---|---|---|---|
| | DRO | ARL | VFair | DRO | ARL | VFair |
| UCI Adult | 0.2956 | 0.9957 | 0.9955 | 32.75% | 97.45% | 96.55% |
| Law School | 0.1693 | 0.6106 | 0.5839 | 37.19% | 95.61% | 95.91% |
| CelebA | 0.1663 | 0.1886 | 0.1474 | 66.47% | 95.41% | 94.62% |

As evidenced by Law School and CelebA, similar predictions do not necessarily indicate similar model parameters. Note that VFair could be more distinct from ERM compared to other fair models, especially on complicated image dataset CelebA. The better fairness improvement in Table 5 also proves that VFair can explore different minima in broader model space, guiding the model to converge to a fairer point.

## F.5 Regression results on UCI Adult

Table 9: Comparison of regression results ($\times 10^2$) on UCI Adult. Here, ↓ is for Utility and WU because MSE is used, and smaller values indicate better utility.

| | | Utility ↓ | WU ↓ | MUD ↓ | TUD ↓ | VAR ↓ |
|---|---|---|---|---|---|---|
| UCI Adult | ERM | $\mathbf{11.37}_{(0.77)}$ | $14.34_{(1.13)}$ | $10.83_{(1.16)}$ | $17.72_{(2.00)}$ | $4.94_{(1.01)}$ |
| | DRO | $22.54_{(0.78)}$ | $23.13_{(0.57)}$ | $2.12_{(0.87)}$ | $3.54_{(1.4)}$ | $0.12_{(0.06)}$ |
| | ARL | $11.70_{(0.39)}$ | $14.78_{(0.49)}$ | $11.23_{(0.54)}$ | $18.26_{(0.82)}$ | $4.63_{(0.69)}$ |
| | MPFR | - | - | - | - | - |
| | FKL | - | - | - | - | - |
| | VFair | $11.49_{(0.27)}$ | $\mathbf{14.24}_{(0.34)}$ | $\mathbf{9.92}_{(0.39)}$ | $\mathbf{16.25}_{(0.73)}$ | $2.52_{(0.43)}$ |
| | Improved | -0.12 | +0.1 | +0.91 | +1.47 | +2.42 |

Table 9 shows the compared regression results on the UCI Adult dataset. Note that MPFR and FKL are not designed with stochastic updates and they suffer from out-of-memory issues under our experimental setup on the UCI Adult dataset.

### F.6 Comparison with Rawlsian fair classification methods with sensitive attributes

Table 10: Comparison with methods with access to attributes, where the utility-based results are with %, and the results of VAR are $\times 10^2$ for a neat presentation.

|  |  | Utility ↑ | WU ↑ | MUD ↓ | TUD ↓ | VAR ↓ |
|---|---|---|---|---|---|---|
| UCI Adult | PMG | 79.48 | 73.02 | 22.36 | 27.85 | 416.86 |
|  | MMPF | **85.33** | **81.23** | **12.78** | **14.38** | - |
|  | VFair | 84.74 | 80.36 | 15.71 | 20.71 | 8.17 |
| Law School | PMG | 78.52 | 70.14 | **9.47** | **15.59** | 81.83 |
|  | MMPF | 82.72 | 74.70 | 9.92 | 18.57 | - |
|  | VFair | **85.40** | **75.25** | 11.00 | 19.91 | 6.29 |
| COMPAS | PMG | 53.52 | 49.14 | 9.97 | 10.58 | 13.21 |
|  | MMPF | 66.39 | **63.91** | **2.15** | **5.44** | - |
|  | VFair | **66.80** | 63.86 | 6.25 | 8.47 | 1.86 |

We have further supplemented control experiments, where the model has access to sensitive attributes and is optimized under constrained regularization. In detail, we reproduced the MMPF in [25] and further designed experiments that penalize the losses of the minority group, denoted as PMG. As MMPF is not applicable to image datasets, the results are conducted on three benchmark datasets shown in Table 10. By leveraging additional group information, MMPF achieves improved fairness results, showing that group priors are indeed needed for significant fairness improvement in classification tasks. However, MMPF is not a harmless approach, particularly evident on Law School, where it sacrifices model utility for a fairer point. PMG yields unsatisfactory performance consistently due to its excessive focus on the minority group, missing general information from other groups.

## G  Computational costs

Since the backward pass is the bottleneck of the total computation, we found that VFair requires approximately twice the computation time compared to the ERM method, as shown in Table 11 with the Law School dataset as an example. Note that ARL, an adversarial method, requires a comparable wall-clock time to VFair due to its inner and outer optimization nature.

Table 11: Comparison of four methods' wall-clock time on Law School with the same experimental setup.

| Method | ERM | DRO | ARL | VFair |
|---|---|---|---|---|
| Time | 349.4s | 243.5s | 640.1s | 677.6s |

