# OpenReview forum: "Towards Harmless Rawlsian Fairness Regardless of Demographic Prior"
_NeurIPS.cc/2024/Conference — NeurIPS 2024 poster_

### Official Review · Reviewer_UBKm · 2024-07-09

**Soundness:** 1
**Presentation:** 2
**Contribution:** 2
**Rating:** 4
**Confidence:** 4

**Summary:**

Although importance of group fairness appreciated, most existing works have required demographic information for debiasing. The paper introduces a novel method, VFair, to achieve fairness with minimum sacrifice of the utility under no prior demographic information scenario. VFair aims to achieve harmless Rawlsian fairness via minimizing variance of losses, which is pertinent to disparity of utility across subgroups.

**Strengths:**

1. The paper is easy to read.
2. Considering important topic.

**Weaknesses:**

1. Comparison in the experimental results is not comprehensive.
2. Motivation is less highlighted. It is unclear why the proposed method is unique and how it addresses limitations of previous methods.

Also, please see the question section below.

**Questions:**

- Why does the paper referring to Dirac delta distribution instead of uniform distribution when demonstrating Figure 1? If the indices (x-axis) are sorted by loss value, wouldn’t a uniform distribution be more appropriate for an optimal scenario?
- The paper suggests that MUD (Maximum Utility Disparity) or accuracy parity across groups are implicitly connected to a uniform or Dirac delta distribution of instance-wise losses. Could the authors clarify this connection? Intuitively, these concepts do not seem necessarily interchangeable.
- (line:137-141) The paper mentions overfitting and low variance at the same time, which seems counter-intuitive. Could the authors clarify how they propose to mitigate this apparent conflict?
- How is the proposed method of “harmless fairness” differentiated from the previous work by Zhang et al. (2018)[1] on mitigating unwanted biases with adversarial learning?
- The empirical results do not perfectly align with the TUD (Total Utility Disparity) and variance metrics presented. Could the authors explain these discrepancies?
- There should be a comparison with more state-of-the-art methods to provide a comprehensive evaluation. Could the authors include or discuss more recent and relevant methods in their comparison?


[1] B. H. Zhang, B. Lemoine, and M. Mitchell, “Mitigating Unwanted Biases with Adversarial Learning,” AAAI/ACM Conference on Artificial Intelligence, Ethics, and Society, 2018.

**Limitations:**

Please see questions and weakness parts.

---

> ### Author Rebuttal · Authors · 2024-08-07
>
> # Reply to weakness
> 1. Experimental results. We compared the regression and classification performance among six benchmark methods across six datasets. Additionally, we discussed the results of using F1-score as utility metric, randomly splitting groups, methods using demographic prior during the training phase, and different $lambda$ settings. By thoroughly examining various scenarios and conditions, we have ensured that our analysis and conclusions are well-founded and credible.
>
> 2. Novelty and effectiveness. We studied Rawlsian harmless fairness without demographic information in both classification and regression tasks, which has not been studied before. Moreover, as supported by reviewer kaxC in the 'Strengths' section, our idea is incredibly novel and impressive. Besides one weakly related reference, they were unable to find occurrences of this idea in prior work.
>
> # Reply to questions
>
> 1. Explanation for Dirac delta distribution. Please refer to Figure 1 in the newly uploaded supplementary PDF, where we provide a more detailed description of the Dirac delta distribution. We speculate that your misunderstanding may stem from confusion regarding the meaning of the x-axis of Figure 1 in our paper. The x-axis represents the index of sorted losses, rather than the actual loss values, which is different from Figure 1 in the supplementary PDF.
>
> 2. From MUD to loss distribution. (1) MUD is a group-level metric. When demographic information is inaccessible, we strive to minimize MUD across all possible group divisions. (2) From the perspective of individual instances, when the loss for each example is as similar as possible, MUD will approximate zero regardless of the group division. (3) If the loss for each example approximates zero, the loss distribution will resemble a Dirac delta distribution. If the loss for each example approximates $\mu$ (e.g. $\mu$=0.25 in a 0-1 regression task), the model will behave like a uniform regressor or classifier. Note that this does not imply that the loss distribution itself is uniform; rather, the model's performance is uniformly poor across instances.
>
> 3. Confusion caused by the term 'overfit'. Sorry for any confusion on this point. This section was intended to introduce the derivation of the instance-level loss, which bypasses the unobserved sensitive attributes, rather than implying that our model would overfit. The logic is as follows: (1) We aim for MUD=0 for fairness. (2) Oracle model achieves zero loss for each sample, and thus MUD=0. (3) A loss of 0 on the training set indicates overfitting risks in practice.
>
>    And aiming for MUD=0 does not necessarily lead to overfitting in our method.
>
> 4. Difference with [1]. We study different problems. They focus on achieving fairness with access to demographic information. However, our task is more challenging as we aim to achieve fairness without demographic information.
>
> 5. Consistency of TUD and VAR in experimental results. In our experimental results, TUD and VAR consistently outperform other methods in most cases, except for some classification cases. As discussed in lines 298-301, under discrete utility metrics, even if our method has smaller loss disparities at the instance level, fairness metrics may still not improve.
>
> 6. Comparison with SOTA. Our method uniquely addresses Rawlsian harmless fairness without requiring demographic information in both classification and regression tasks, a topic that has not been extensively studied before. Existing works not included in this comparison generally struggle to adapt to this specific setting. Besides the employed baselines (which were carefully modified for harmless fairness), to the best of our knowledge, no other recent work targets this same research problem.
>
> [1] B. H. Zhang, B. Lemoine, and M. Mitchell, “Mitigating Unwanted Biases with Adversarial Learning,” AAAI/ACM Conference on Artificial Intelligence, Ethics, and Society, 2018.

---

> > ### Comment · Reviewer_UBKm · 2024-08-14
> >
> > Thanks for the detailed explanation. However, I am still conservative about its novelty. So I updated my score accordingly.

---

> > > ### Author Response · Authors · 2024-08-14
> > >
> > > Thank you very much for taking the time to review our manuscript and provide your valuable feedback. However, we would like to reiterate the novelty of our approach:
> > >
> > > 1. **Novel Problem**: We highlight the setting of harmless Rawlsian fairness regardless of demographic prior in both classification and regression tasks. Previous works either more or less require direct or implicit demographic information (e.g., FairRF[9]), do not meet the harmless requirement (e.g., DRO[13]), or can only be applied to classification or regression tasks (e.g., MPFR[15], FKL[36]).
> > >
> > > 2. **Novel Fairness Proxy**: As recognized by reviewer kaxC in the “Strengths” section, we propose minimizing the variance of prediction losses as a straightforward yet effective fairness proxy. To emphasize its novelty, in lines 169-173, we show the differences between our method and variance-bias research[27,28].
> > >
> > > 3. **Novel Update Approach**: We have developed a novel dynamic approach for conducting harmless updates, which operates at both the loss and gradient levels. Compared to ordinary bi-objective optimization approaches[26], our approach further designs $\lambda_2$ from the sample-reweighting perspective to guarantee the weight of each training example is non-negative, making our method connect with the up-weighting concept used in recent worst-case fairness methods[13,12,21].
> > >
> > > 4. **Novel Analysis**: For the first time, we challenge and validate the necessity of the group ratio in both classification and regression problems. Our analysis highlights that regardless of any prior, harmless Rawlsian fairness is achievable in regression tasks but not in classification tasks. As shown in Fig. 3, due to the discrete metric and unchanged group partition, the accuracy-based metrics' values remain unchanged even with a smaller sample disparity. Therefore, the improvement in fairness in classification tasks is still bound by the overall utility. However, regression problems using a discrete metric are not limited by this and can achieve significant fairness improvement under our setting.
> > >
> > > We hope this clarifies the novelty and significance of our contributions. We are always willing to address any further questions you may have.

---

### Official Review · Reviewer_kaxC · 2024-07-11

**Soundness:** 2
**Presentation:** 3
**Contribution:** 3
**Rating:** 7
**Confidence:** 4

**Summary:**

The authors propose VFair: an approach to Rawlsian, demographics-agnostic fairness where the variance over each data point's loss term is minimized together with the mean loss during training. These objectives are clearly often at odds with eachother, so they include a principled dynamic weighting scheme for the multi-objective optimization, making use of the mathematical relationship between the sample mean and variance. In experiments, this appears to outperform other baselines that are agnostic to demographics information.

**Strengths:**

The authors effectively propose an incredibly simple idea: instead of only minimizing the mean loss, we also minimize the sample variance over all individual loss terms. It strikes me as an idea that must have been investigated before, but besides one weakly related references that were missed (Spady and Stouli, 2018), I was unable to find occurrences of this idea in prior work.

The dynamic weighting for the objectives is intriguing, as it combines a lower bound based on black-box optimization (apparently not original), with a second, more stable lower bound that is well-motivated by exploiting the relation between the sample variance and mean.


Spady, Richard, and Sami Stouli. "Simultaneous mean-variance regression." arXiv preprint arXiv:1804.01631 (2018).

**Weaknesses:**

W1. The motivation for using the variance of as a second objective is found in Prop. 1, which reads "For any s that splits data into a number of groups, u ⊥ s holds if and only if the loss ℓ is (approximately) independent of the training example z, i.e., ℓ ⊥ z.". The first condition is either very unclearly phrased or simply incorrect. It seems to say that *for any split*: it holds that u ⊥ s iff ℓ ⊥ z. This clearly cannot be true: the mean accuracies of all groups can be equal while the individual losses within the groups can vary. Instead, Prop. 1 should read "u ⊥ s holds for any s that splits data into a number of groups, if and only if the loss ℓ is (approximately) independent of the training example z, i.e., ℓ ⊥ z.".

Also, Prop. 1 only holds if $u$ is a fully decomposable sum over all individual data samples (just like ℓ is).

W2. The extra motivation for minimizing the variance given in L169-L180 lacks rigour. First, Bennett's inequality uses the actual variance and not the sample variance (which VFair is optimizing), and it is unclear how they are related in the bound. Second, why care about this bound? We care about the actual mean, but not about so much about the gap between the actual and the empirical mean.

W3. Related to W1, I more broadly wonder whether this approach even fits in the popular ML fairness literature. The idea of formalizing discrimination as a problematic bias in ML is that there is some unethical pattern in whom is being disadvantaged by an algorithm. If we are looking at the utility for each data point individually, disregarding demographics, are we still talking about an approach that "does not require demographic information to be fair"? It seems like we are now just saying that we don't *care* about the demographics. This doesn't make the proposed definition of fairness uninteresting, but it does take away the societal motivation for this work (in relation to discrimination law).

**Questions:**

Q1. Could you explain why the 'training losses exhibit a Dirac delta distribution'? Because you are stripping the variance?

Q2. The derivation of Eq. 8 can be more intuitively explained: the Z-score of a random variable $\geq 0$ is always lower-bounded by $-\frac{\mu}{\sigma}$, so $\lambda$ must be lower-bounded by $\frac{\mu}{\sigma}$ for $w_i$ to be positive.

**Limitations:**

No, for societal impact, please refer to papers that more broadly discuss the problems with a technical approach to algorithmic fairness and the problematic assumptions we need to make (e.g. relating to the measurability of the utility).

The limitations were not discussed in detail.

---

> ### Author Rebuttal · Authors · 2024-08-07
>
> # Reply to weakness
>
> 1. Rigor of expression. Thank you for your careful reading and rigorous derivation. We will update Proposition 1 and the assumptions regarding u as you suggested.
>
> 2. Confusion caused by lines 169-180. Sorry for any confusion on this point. Lines 169-180 offer a deeper reflection on our method and do not serve as extra motivation. After deriving Equation (2), we further considered the commonalities and differences between our method and existing works. Equation (3) illustrates the fundamental difference between our method and bias-variance research, while Equation (4) demonstrates the essential similarity between our method and DRO.
>
> 3. About demographic prior. We believe that demographic prior is meaningful, for demographic prior is still used during the evaluation stage. Table 7 shows a comparison between VFair and methods using demographic prior during the training stage. Results indicate that demographic prior does indeed provide some improvement.
>
> # Reply to questions
>
> 1. Explanation for Dirac delta distribution. Please refer to Figure 1 in the newly uploaded supplementary PDF, where we provide a more detailed description of the Dirac delta distribution.
>
> 2. More concise explanation. Thank you for your suggestion. We will improve the clarity and logic of Remark 2 following your comments.
>
> # Reply to limitations
>
> We will add a Limitations section to discuss the additional computational costs of the proposed method. Briefly, our method, VFair, requires two backward propagations, resulting in approximately double the computation time compared to ERM. Please also refer to our detailed responses to reviewer cPfu.

---

> > ### Comment · Reviewer_kaxC · 2024-08-12
> >
> > Thanks for the clarifications. I read the other reviewers' comments and still really like the paper, so I'll continue to advocate for its acceptance (with my current score).

---

> > > ### Author Response · Authors · 2024-08-13
> > >
> > > Thank you so much for your recognition of our work. Once again, we sincerely appreciate the time and effort you spent reviewing our paper. Your valuable suggestions and insights have greatly contributed to improving our manuscript.

---

### Official Review · Reviewer_Xh7a · 2024-07-12

**Soundness:** 2
**Presentation:** 2
**Contribution:** 3
**Rating:** 5
**Confidence:** 3

**Summary:**

The authors suggest a framework aimed at enhancing the fairness guarantees of classifiers in scenarios where sensitive information is unavailable. Their approach seeks to identify a classification rule that minimizes the variance in losses from the training sample, while ensuring that the overall average utility does not significantly decline from the maximum possible utility.

**Strengths:**

S1 - The paper successfully validates all the claims outlined in the abstract and introduction. Each assertion is supported with empirical evidence and detailed analysis. The authors have meticulously followed through on their initial promises, providing a cohesive and comprehensive study that aligns well with the stated objectives.

S2 - The problem addressed in this paper is of high relevance in reality. Ensuring the fairness of classifiers, especially in the absence of sensitive information, is a crucial issue with significant practical implications.

S3 - The authors effectively address the issues and challenges posed by the general methodology when applied in practical scenarios. For instance, the insightful analysis on gradient alignment enhances the practical applicability of their approach.

S4 - This analysis offers valuable insight into an important finding in the field of algorithmic fairness: the distinctions between achieving fairness in classification and regression tasks.

S5 - The abstract and the introduction effectively substantiate all the claims made, including the contributions put forth by the authors. These assertions find validation through the description of the methodology employed and the experiments conducted. The method section elaborates on the techniques and approaches considered, demonstrating how they align with the stated objectives. Furthermore, the experimental results provide empirical evidence that supports the claims made in the introduction.

**Weaknesses:**

W1 - Reproducibility issues. The experimental setting lacks detail regarding critical aspects such as the employed learning rate, the number of epochs, and the hyperparameters or tuning procedures for different methods. Additionally, the authors do not report how the dataset is split. Providing this information is essential for enabling others to replicate the study and verify its findings.

W2 - The paper overlooks an important body of work from the field of algorithmic fairness that operates without knowledge of demographic information: [1]. This work constitutes a significant contribution to the field and could provide valuable context and support for the current study. Besides, including and considering such an approach in the experimental section would enhance the quality and informativeness of the provided results and derived conclusions.

W3 - The proposed fairness criteria operate at the instance level, thus can be viewed as an individual fairness strategy. However, the authors do not highlight this similarity or discuss their approach in relation to other existing works based on individual fairness. It would be beneficial to include this information, reference these works, and incorporate some comparisons in the experimental section.

W4 - The discussion of the results includes several overstatements. For example, on lines 307-310, the authors claim that VFair yields superior performance. However, this is an exaggeration. While VFair may exhibit higher utility in some cases, it comes with a substantial cost to fairness. Moreover, even when VFair demonstrates higher overall utility, the improvement is marginal, less than 1%, which does not substantiate the claim of superiority.

W5 - The paper lacks insight into the computational cost of the proposed method. There is no empirical validation to determine whether it is more costly than existing methods and, if so, to what extent. Providing this information would be valuable, as it would illuminate whether a trade-off exists between computational cost and the results obtained without demographic information. Understanding this balance is crucial for assessing the practical applicability and efficiency of the proposed approach.

W6 - The authors claim in line 256 that DRO needs the identification of the worst-off group through a bound of group ratio, but as far as I am aware the latter is not true. The method only requires defining the value of $\eta$ but has nothing to do with the demographic information.


W (minor) - The section titles should only contain upper case letters at the start of the first word, and not at the beginning of every word in the title.

W (minor) - In equation 5 you employ the parameter $\eta$, which is also used in equation 4 but to refer to a completely different parameter. I would recommend using different letters for each of the parameters to avoid confusion.



[1] Martinez, N. L., Bertran, M. A., Papadaki, A., Rodrigues, M., & Sapiro, G. (2021, July). Blind pareto fairness and subgroup robustness. In International Conference on Machine Learning (pp. 7492-7501). PMLR.

**Questions:**

Q1 - Why don't you bold the DRO results in Table 1 when they outperform the results of your proposed method?

Q2 - What are the Z-scores?

Q3 - How well does it scale with the increasing number of instances?

Q4 - In line 141 the authors talk about the risk of overfitting, however they do not provide a deep insight into the issue. How likely is it that it happens? Under which circumstances?

Q5 - What happens to the method when outlier instances are present? These outliers could significantly complicate the optimization of the classification rule or even result in a trivial classification rule.

**Limitations:**

The authors do not clearly state the limitations of their proposed method.

As a suggestion, it would be interesting to address potential issues such as overfitting, scalability, and susceptibility to outliers.

---

> ### Author Rebuttal · Authors · 2024-08-07
>
> # Reply to weakness
> 1. Reproducibility issues. We will provide more training details in the appendix. (1) Throughout experiments, all methods are set with a batch size of 32 and a learning rate of 0.01. As for the training epoch, as mentioned in lines 555-559, to ensure all baselines comply with the harmless fairness setting, we terminate each training process at the epoch with the nearest loss value to a well-converged ERM. (2) The datasets are randomly split in a 7:3 ratio following ARL's experimental implementation.
> 2. Comparison with BPF. We have discussed the relationship between our method and BPF, referenced as [12] in our paper. BPF, as an improved version of DRO, also utilizes group ratio prior. Considering your comments, we added BPF as another baseline. However, as shown in Table 1 in the newly uploaded supplementary PDF, VFair still outperforms BPF on all metrics.
> 3. Connection with Individual Fairness. Although VFair operates at the instance level, it differs from Individual Fairness. (1) Individual Fairness requires that if two individuals are close on the similarity metric, they should be close on the treatment metric, resulting in a continuous treatment metric outcome. (2) VFair falls in (Rawlsian) group fairness, which does not consider the similarity between individuals but focuses on performance disparities between different groups, pursuing close treatment (utility in this paper) metric outcomes. (3) As mentioned in line 145, group fairness pursues $\ell\perp z$, while Individual Fairness pursues $\ell\sim f(z)$, where $f()$ represents a similarity metric. (4) The similarity metric $f()$ also serve as a form of prior. In contrast, VFair does not call for any form of prior.
> 4. Analysis of results. We have strived to present our findings accurately and objectively and we believe there is no exaggeration of effects. Note that we study harmless fairness; therefore, only methods that do not compromise much on overall utility will be finally ranked. (1) Regarding Table 2, DRO turns out a uniform regressor with low utility. In this context, our method outperforms other methods on all metrics except for DRO. (2) Note that lines 307-310 mention that Table 2 aims to show the limited improvement on UCI but significant improvement on CelebA (except for DRO). We acknowledged the limited improvement on UCI and analyzed the reason.
> 5. Computational costs. Thank you for this suggestion. (1) Our proposed method requires two rounds of back-propagation, which thus leads to more computation cost compared to ERM or DRO. (2) Since the backward pass is the bottleneck of the total computation, we found that VFair requires approximately twice the computation time compared to the ERM method, as shown in Table 1 with the Law School dataset as an example. (3) Note that ARL, an adversarial method, requires a comparable wall-clock time to VFair due to its inner and outer optimization nature.
>
>    Table 1: Comparison of four methods' wall-clock time on Law School with the same experimental setup.
>    |Method|Time|
>    |-|-|
>    |ERM|349.4s|
>    |DRO|243.5s|
>    |ARL|640.1s|
>    |VFair|677.6s|
> 6. Eta in DRO. When applying DRO, using a defined eta can be interpreted as using some group ratio. As evidenced in the official code of DRO in dual_robust_opt.py, the calculation of eta is achieved by bi-search with group ratio eps as input of the get_rho() function.
> 7. Minor weakness. Thank you for this suggestion. We will edit the section title and the symbol expressions according to your advice.
> # Reply to questions
> 1. Presentation of DRO results. Similar to W4, we study harmless fairness. As mentioned in lines 270-273 and lines 65-68, DRO turns out a uniform regressor with low utility.
> 2. Explanation of Z-Score. We use Z-score as it is a fundamental statistical measure. It is calculated by subtracting the mean from the individual value and then dividing it by the standard deviation. In our context, the Z-score of each example's loss is calculated as  $\frac{\ell - \hat{\mu}}{\hat{\sigma}}$.
> 3. Adaptation on larger datasets. As shown in Appendix D, our method adopts a stochastic optimization strategy, from which one can see the complexity will be linear to the number of training samples. Note that baseline methods MPFR and FKL are not designed with stochastic updates and suffer from out-of-memory issues, as shown in Table 6.
> 4. Confusion caused by the term 'overfit'. Sorry for any confusion on this point. This section was intended to introduce the derivation of the instance-level loss, which bypasses the unobserved sensitive attributes, rather than implying that our model would overfit. The logic is as follows: (1) We aim for MUD=0 for fairness. (2) Oracle model achieves zero loss for each sample, and thus MUD=0. (3) A loss of 0 on the training set indicates overfitting in practice.
>
>    Aiming for MUD=0 does not necessarily lead to overfitting.
> 5. Discussion about outliers. (1) We agree that the outlier problem has been studied in Fairness problem. We have also investigated some works in this direction, such as DORO[1], ARL, and GRASP[2]. However, these methods have different strategies to solve the outlier problem, and there is no general, representative method. (2) According to our analysis in the Loss view on page 5, we can simply cap the value of Z-score, preventing the model from excessively concentrating on outliers that produce large losses. As shown in Table 5, on the COMPAS dataset whose labels are found to be noisy [3], VFair achieves the best performance (except for FairRF which leverages feature correlations prior). We will leave a thorough comparison with other existing strategies in future work.
> # Reply to limitations
> Refer to the above answers.
>
> [1] Zhai R, et al. Doro: Distributional and outlier robust optimization.
>
> [2] Y. Zeng, et al. Outlier-robust group inference via gradient space clustering.
>
> [3] Preethi L, et al. Fairness without demographics through adversarially reweighted learning.

---

### Official Review · Reviewer_cPfu · 2024-07-20

**Soundness:** 3
**Presentation:** 4
**Contribution:** 3
**Rating:** 8
**Confidence:** 5

**Summary:**

The paper proposes a novel view of Rawlsian fairness for scenarios where no demographic information is provided. The core proposal of the paper is VFair, a method for reducing the variance of the predictive loss across a dataset, with the core tenet that a well-concentrated loss distribution would assign similarly-beneficial outcomes to all participants. They also discuss connections and differences with exisiting 'worst-case' approaches such as Distributionally Robust Optimization and Blind Pareto Fairness.

**Strengths:**

The motivation behind the proposed VFair is easily understood as a constrainted optimization objective of minimizing predictive loss variance subject to a performance constraint. This is implemented via a simple Lagrange multiplier approach which essentially weights the standard loss minimization objective with an additional (empirical) loss variance objective .

The multiplier itself is actually not optimized for exactly, rather its lower bounded to ensure two reasonable conditions on the update:
A) The overall loss gradient points towards (mean) loss reduction (i.e., the variance reduction component does not overpower the mean loss reduction component)
B) The per-sample effective loss is positively weighted (i.e., no sample is encouraged to increase its loss value)

The resulting method relies on a simple exponential moving average of the overall predictive loss of the dataset and a few simple computations.

**Weaknesses:**

my main concerns with this paper are the following:

A) Computational costs. Steps 8/9 in Algorithm 1 require the independent computation of the gradient wrt the mean objective and the gradient wrt the std objective. While not egregious, this would require two backwards passes through the network so the method is computationally more intensive than some of the discussed alternatives like DRO-BPF

B) Although the motivation of VFair is shown in Eq 1, the loss constraint $E_z[\ell(z,\theta)]\le \delta$ (in particular the delta parameter) does not end up playing a role in the final method. It is not immediately apparent what the equivalent delta value would be for the proposed method

**Questions:**

See point B in weaknesses

**Limitations:**

Yes

---

> ### Author Rebuttal · Authors · 2024-08-07
>
> # Reply to weaknesses
>
> 1. Computational costs. Your comment on two backward passes is correct. We will include this extra computation cost as one of the limitations of our work. Since the backward pass is the bottleneck of the total computation, we found that VFair requires approximately twice the computation time compared to the ERM method, as shown in Table 1 with the Law School dataset as an example. Note that ARL, an adversarial method, requires a comparable wall-clock time to VFair due to its inner and outer optimization nature.
>
>    Table 1: Comparison of four methods' wall-clock time on Law School with the same experimental setup.
>
>    | Method | Time   |
>    | ------ | ------ |
>    | ERM    | 349.4s |
>    | DRO    | 243.5s |
>    | ARL    | 640.1s |
>    | VFair  | 677.6s |
>
> 2. Delta value. Sorry for any confusion on this point. (1) When presenting the optimization steps in the paper, we considered VFair to be executed independently, so we might not know the appropriate delta value beforehand. In this sense, the final delta value a trained model can achieve will be mainly determined by the maximal number of epochs, assuming it is well-converged. (2) In the experiments, to ensure all baselines comply with the harmless fairness setting, we terminate each training process at the epoch with the nearest loss to the delta derived from a converged ERM (See lines 555-559).

---

> > ### Comment · Reviewer_cPfu · 2024-08-12
> >
> > Thank you for the response. I will keep my score

---

> > > ### Author Response · Authors · 2024-08-13
> > >
> > > Thank you so much for your recognition of our work. Once again, we sincerely appreciate the time and effort you spent reviewing our paper. Your valuable suggestions and insights have greatly contributed to improving our manuscript.

---

### Author Rebuttal · Authors · 2024-08-07

We appreciate the valuable comments from the reviewers. In response to some issues that required additional experiments and illustrations, we added new experiments and figures in the supplementary PDF.

---

### Comment · Area_Chair_7vHC · 2024-08-11
**Encouraging discussion**

Dear reviewers,
The authors have responded to your questions. Can you please check them and update your evaluation of this work as needed?
Many thanks!

-AC

---

### Decision · Program_Chairs · 2024-09-25

**Decision:**

Accept (poster)

**Comment:**

This paper proposes an interesting approach to achieving Rawlsian fairness in scenarios where no demographic information is provided. The paper's novel approach was appreciated, and the reviewers and the AC also welcomed its theoretical rigor.  My recommendation is to accept it. Should the paper make the cut, however, the final version should include a more detailed discussion of related work, explicit information on the experimental setup, and a clearer explanation of the computational costs.